# Variants in *NR6A1* cause a novel oculo vertebral renal syndrome

Uma M. Neelathi [1,10], Ehsan Ullah [1,10], Aman George [1], Mara I. Maftei[2], Elangovan Boobalan [1], Daniel Sanchez-Mendoza[1], Chloe Adams [1], David McGaughey[1], Yuri V. Sergeev [1], Ranya AI Rawi[1], Amelia Naik [1], Chelsea Bender[1], Irene H. Maumenee[3], Michel Michaelides[2,4], Tun Giap Tan [5], Siying Lin [2,4], Rafael Villasmil[6], Delphine Blain [1], Robert B. Hufnagel[1,7], Gavin Arno [2,8], Rodrigo M. Young [2,9,11], Bin Guan [1,11] & Brian P. Brooks [1,11] ✉

Colobomatous microphthalmia is a potentially blinding congenital ocular malformation that can present either in isolation or together with other syndromic features. Despite a strong genetic component to disease, many cases lack a molecular diagnosis. We describe an autosomal dominant oculo-vertebral-renal (OVR) syndrome in six independent families characterized by colobomatous microphthalmia, missing vertebrae and congenital kidney abnormalities. Genome sequencing identified six rare variants in the orphan nuclear receptor gene *NR6A1* in these families. We performed in silico, cellular, and zebrafish experiments to demonstrate the *NR6A1* variants were pathogenic or likely pathogenic for OVR syndrome. Knockdown of either or both zebrafish paralogs of *NR6A1* results in abnormal eye, kidney, and somite development, which was rescued by wild-type but not variant *NR6A1* mRNA. Illustrating the power of genomic ascertainment in medicine, our study establishes *NR6A1* as a critical factor in eye, kidney, and vertebral development, and a pleiotropic gene responsible for OVR syndrome.

Uveal coloboma is a congenital ocular malformation caused by failure of the ventral optic fissure to close during early eye morphogenesis and is usually considered on a phenotypic continuum with microphthalmia and anophthalmia[1–5]. A rare condition[6–11], coloboma may nonetheless account for up to 10% of childhood blindness[12]. Although significant progress has been made in identifying genes associated with syndromic and non-syndromic coloboma, the yield of diagnostic testing remains low, especially for isolated, non-syndromic coloboma, suggesting other genes are yet to be discovered[13–15]. To identify novel coloboma genes, the National Eye Institute has conducted a natural history study since 2006, on the genetics of coloboma that includes systematic deep

phenotyping of probands and first-degree family members. We have previously identified a syndrome characterized by missing vertebrae (in the thoracic and/or lumbar spine), congenital kidney abnormalities, and uveal coloboma, inherited in an autosomal dominant fashion with incomplete penetrance and variable expressivity[16].

We identified structural and sequence variants in the transcription factor gene *NR6A1* (*Nuclear receptor subfamily 6, group A, member 1*, OMIM*602778) in three families by genome sequencing (GS). These results were extended via analysis of the Genomics England 100,000 Genomes Project (UK100KGP), where three additional individuals with microphthalmia/anophthalmia/coloboma were identified[17].

[1]Ophthalmic Genetics & Visual Function Branch, National Eye Institute, National Institutes of Health, Bethesda, MD, USA. [2]UCL Institute of Ophthalmology, University College London, London, UK. [3]Harkness Eye Institute, Columbia University, New York, NY, USA. [4]Moorfields Eye Hospital, NHS Foundation Trust, London, UK. [5]Torbay Hospital, Torbay and South Devon NHS Foundation Trust, Devon, UK. [6]Flow Cytometry Core, National Eye Institute, Bethesda, MD, USA. [7]Center for Integrated Health Care Research, Kaiser Permanente Hawai'i; Hawai'i Permanente Medical Group, Honolulu, HI, USA. [8]Greenwood Genetic Center, Greenwood, SC, USA. [9]Center for Integrative Biology, Universidad Mayor, Santiago, Chile. [10]These authors contributed equally: Uma M. Neelathi, Ehsan Ullah. [11]These authors jointly supervised this work: Rodrigo M. Young, Bin Guan, Brian P. Brooks. ✉e-mail: brooksb@mail.nih.gov

Originally termed germ cell nuclear factor (*GCNF*)/retinoid receptor-related testis-associated receptor (*RTR*), *NR6A1* is an orphan member of the nuclear hormone receptor family of transcription factors, often acting as a transcriptional repressor. *NR6A1* is highly expressed in embryonic and other stem cells from various tissues (especially testes) and is repressed upon differentiation. *NR6A1* plays an important role in somite and subsequent vertebral development in mice, and in livestock species it is correlated with vertebral number[18–21]. To our knowledge, there are no reports on the role of *NR6A1* in eye or kidney development.

Here we described an autosomal dominant oculo-vertebral-renal (OVR) syndrome caused by variants in the orphan nuclear receptor gene *NR6A1*, supporting the pathogenicity of variants through a combination of in silico, in vitro, and in vivo investigations. To our knowledge, this is an undescribed Mendelian trait in humans characterized by missing vertebrae.

## Results

### Variants in NR6A1 cause an oculo-vertebral-renal (OVR) syndrome

We identified three rare *NR6A1* variants in three families affected by uveal coloboma (COL005, COL034, COL171) with or without microphthalmia, cataract, and missing vertebrae through genome sequencing. In cases where multiple generations are affected, transmission is

autosomal dominant with incomplete penetrance and variable expressivity (Fig. 1a). Clinical data for all the participants with a positive molecular result is shown in Table 1. No other candidate pathogenic variants in *NR6A1* were identified in the NEI coloboma cohort consisting of a total of 224 probands (66 analyzed by genome sequencing, 57 by exome sequencing, and 101 by amplicon sequencing).

The proband of the family (COL005.1) presented at age 14, with bilateral uveal colobomas (Fig. 1b, c). Family history was notable for a younger brother (COL005.4), a second cousin (COL005.10), and a first cousin once removed (COL005.17) with uveal coloboma. The deletion breakpoints were in intron 2 and 6 removing the coding sequence for amino acids (aa) Ile48-Gly275 and likely causing a frameshift (p.Ile48Asnfs*3, Fig. 1h). The status of the heterozygous deletion was determined by breakpoint PCR among family members available, which revealed complete segregation with the missing vertebrae with an estimated LOD score of 3.6 (Fig. 1a, Supplementary Fig. 1). Four family members were also affected by coloboma in addition to missing vertebra, of which one (COL005.17) also had unilateral renal agenesis by report. The proband of the family (COL034.1) presented at age 11 months with bilateral uveal colobomas and microphthalmia OS (Fig. 1e). Prenatal history was remarkable for the inability of an ultrasound at 18 weeks to visualize the left kidney. Systemic testing demonstrated 10 thoracic vertebrae and left renal agenesis. Genome sequencing revealed a heterozygous c.274C>T p.(Arg92Trp) variant in

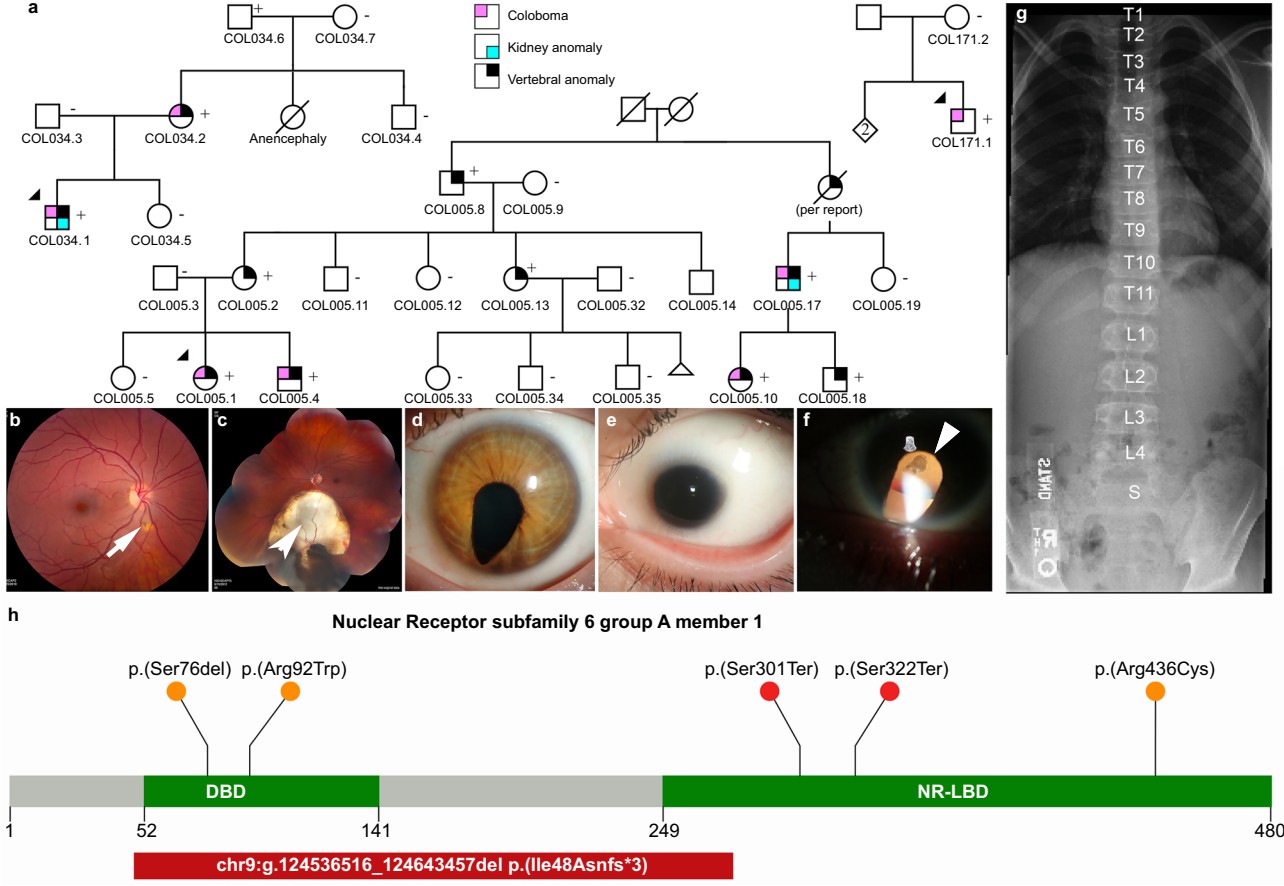

**Fig. 1 | Phenotypes associated with variants in *NR6A1*. a** Pedigrees of three families (COL005; COL034; COL171) from the NEI cohort demonstrating coloboma with or without microphthalmia and cataract, missing vertebrae, and congenital renal anomalies. Inheritance is autosomal dominant with incomplete penetrance and variable expressivity. **b** Linear pigmentary disturbance representing a *forme fruste* of coloboma (arrow) in COL005.1 (right eye). **c** Larger chorioretinal coloboma in the left eye of COL005.1 demonstrating a retinal tear in the far periphery (arrowhead). **d** Iris coloboma of the left eye of COL005.10. **e** Microphthalmia of the

left eye in COL034.1. **f** Retro-illumination image of the left eye of COL171.1 demonstrating iris coloboma and posterior subcapsular cataract (open arrow). **g** Spine x-ray of COL005.4 demonstrating 11 thoracic (normal 12) and 4 lumbar (normal 5) vertebrae. **h** Schematic of NR6A1 variants detected in the NEI and UK Genomics England cohorts. + individual with variant, − individual without variant. DNA binding domain (DBD) and putative nuclear receptor ligand binding domain (NR-LBD) are noted (Q15406; InterPro).

## Table 1 | Phenotypic details of individuals carrying pathogenic variants in the NEI cohort

| Family | Member* | Sex | Iris coloboma | Retinal/choroidal coloboma | Optic nerve coloboma | Other ocular findings | Vertebral findings | Renal findings | Audiology findings | Other positive medical findings | Negative medical findings |
|---|---|---|---|---|---|---|---|---|---|---|---|
| COL005 | COL005.1 | F | OS | OU (OD is forme fruste with a small pigmented area inferior to the disc) | OS (slightly anomalous disc) | Amblyopia requiring patching; strabismus requiring surgery; retinal tear OS requiring laser retinopexy; anterior lens pigment and small lens coloboma OS | 11 thoracic vertebrae, spina bifida occulta at S1 and minimal scoliosis | None | Not done | Small jaw and abnormal dentition requiring minor surgeries; mitral valve prolapse; limited left sided proctitis (inflammatory bowel disease), internal hemorrhoids; abnormal contour of the left globe on brain MRI | Normal brain MRI (aside from left globe), serum vitamin A, bone age, cholesterol levels, thyroid function, karyotype |
|  | COL005.2 | F | None | None | None | Myopia, astigmatism and presbyopia | 11 thoracic vertebrae and cervical spondylosis | Not done | Non-congenital low frequency hearing loss by report | History of unexplained relapsing fevers with rash | Normal thyroid function |
|  | COL005.4 | M | None | OD | OU (OS is forme fruste with an anomalous disk with small inferior crescent) | None | 11 thoracic vertebrae, 4 lumbar vertebrae and minimal scoliosis | None | Not done | No further testing done | Normal brain MRI, serum vitamin A, bone age, cholesterol levels, urinalysis |
|  | COL005.8 | M | None | None | None | BCVA was not corrected to 20/20 | 11 thoracic vertebrae and scoliosis | None | Not done | No further testing done | No further testing done |
|  | COL005.10 | F | OU (OD: forme fruste with focal iris transillumination inferiorly) | OU | OU (anomalous discs) | Posterior subcapsular cataract OS; mild heterochromia; history of increased intraocular pressures, treated with latanoprost; best-corrected visual acuity OS 20/40 | Transitional thoracolumbar and lumbosacral vertebral bodies | None | Notched configuration of audiogram in left ear at 2000Hz, with evidence of a slight air-bone gap at that frequency | Abnormal contour of the globes on brain MRI | Normal brain MRI (aside from globes), serum vitamin A, cholesterol levels, serum chemistries, liver function tests, urinalysis, karyotype |
|  | COL005.13 | F | None | None | None | Myopia | 11 thoracic vertebrae and scoliosis of upper thoracic spine | Not done | Not done | History of hypothyroidism | No further testing done |
|  | COL005.17 | M | Not done | Coloboma (unsp.) per report | Not done | Not done | One missing vertebra per report | Missing kidney per report | Not done | No further testing done | No further testing done |
|  | COL005.18 | M | None | None | None | Myopia | 11 thoracic vertebrae | Not done | Not done | No further testing done | No further testing done |
| COL034 | COL034.1 | M | OS | OU | OU | Microphthalmia OS; microcornea ou with pigment deposition on endothelium OS; shallow anterior chamber OS with elevated intraocular pressure; inferonasal cortical cataract OD; moderate amplitude, low frequency nystagmus with an anomalous head posture | 10 thoracic vertebrae | Missing left kidney | None | Small size of the optic chiasm and optic nerves, and abnormal contour of the globes on brain MRI; curvature to superior prominence to the ears; history of left undescended testicle; history of growth hormone deficiency and hypothyroidism | Normal chromosomal microarray; normal pituitary gland on pituitary gland MRI; normal echocardiogram |
|  | COL034.2 | F | None | OS (retinal thinning and pigment inferior to disc) | None | Non-visually significant anterior subcapsular cataracts and anterior pigment OU | 11 thoracic vertebrae and spina bifida occulta at L4, L5, and S1 | None | Not done | None | Normal serum chemistries |

**Table 1 (continued) | Phenotypic details of individuals carrying pathogenic variants in the NEI cohort**

| Family | Member* | Sex | Iris coloboma | Retinal/choroidal coloboma | Optic nerve coloboma | Other ocular findings | Vertebral findings | Renal findings | Audiology findings | Other positive medical findings | Negative medical findings |
|---|---|---|---|---|---|---|---|---|---|---|---|
| | COL034.6 | M | None | None | OU (slightly dysplastic and borderline small optic nerves) | Early stage nuclear and cortical cataracts OU; midperipheral temporal patchy deep depigmentation OD, possibly from old infection | Minimal disc space narrowing noted posteriorly at C5-6 with small anterior soft tissue ossification at C5-6, C6-7; minimal degenerative changes in the lumbar spine | Not done | Not done | History of parathyroid surgery; hypercholesterolemia | No further testing done |
| COL171 | COL171.1 | M | OU | OU | OU | Nystagmus and mild abduction deficits; bilateral microcornea; bilateral posterior subcapsular and nuclear cataracts and missing zonules inferiorly | Mild anterior longitudinal ligament calcification at C5-6 and C6-7 and possible vascular calcification noted within the soft tissues anterior to the L4-5 disc level | Not done | Mild to moderate sensorineural hearing loss across all frequencies in the right eye and mild sensorineural hearing loss in high frequencies of the left ear | Mild elevation of liver function tests; mild elevation of calcium level on mineral panel | Normal urinalysis |

*Subject numbers are noted in the pedigrees in Fig. 1

*NR6A1*, which was found in the affected mother and unaffected grandfather (Fig. 1a, h). The proband of the family (COL171.1) presented at age 36 with bilateral colobomatous microphthalmia affecting the iris, retina/choroid, and optic nerve. Slit lamp exam was notable for bilateral microcornea, bilateral posterior subcapsular and nuclear cataracts, and missing zonules inferiorly OU (Fig. 1f). Genome sequencing revealed a heterozygous c.1306C>T p.(Arg436Cys) variant in the proband which was absent in his unaffected mother (Fig. 1h). Detailed study of the probands and their family members available for evaluation, were described in clinical vignettes in the Supplementary Notes. No convincing pathogenic variants in known coloboma genes were identified in any of these subjects.

### Genome-first approach for NR6A1 variants corroborates microphthalmia, anophthalmia, coloboma (MAC) phenotypes

We performed an unbiased disease association analysis of rare pLoF variants using the UK100KGP dataset[17]. After removing variants resulting from calling artifacts or mis-annotation, only three pLoF variants were found in the cohort with approximately 126,700 alleles (Supplementary Data 1, Supplementary Fig. 2). We found three probands, Proband (A1) presented at age 30, with bilateral chorioretinal coloboma (*forme fruste* OD) and OS coloboma of the optic nerve (Supplementary Fig. 2b). Genome sequencing revealed a heterozygous c.965_980del p.(Ser322Ter), present in both the proband and her unaffected father. Proband (B1) presented at the age of 29, with a severe form of bilateral microphthalmia with a vestigial remnant of eyes, delayed motor development, intellectual disability, abnormal behavior, and schwannoma. This proband carried a heterozygous c.902G>A p.(Trp301Ter) variant. These two nonsense variants are expected to cause loss of protein function either through nonsense-mediated decay or truncation of the putative nuclear receptor ligand binding domain (NR-LBD, Fig. 1h). Proband D (Supplementary Data 1) had a disorder of sex development, carried variant c.288dup p.(Cys96TrpfsTer4), which was absent in either parent. His father was also affected with a disorder of sex development, suggesting that the *NR6A1* variant is likely not associated with the condition.

The UK100KGP MAC cohort, which consists of 215 probands, was queried for rare missense and in-frame insertion/deletion variants. Proband C1, presented at the age of 25 with bilateral microcornea and coloboma affecting the iris, choroid/retina, and optic nerve. One brother had a similar condition by report. Both parents and the two other siblings of the proband had no history of coloboma by report. Genome sequencing revealed a heterozygous variant c.227_229del p.(Ser76del) present in the proband (Fig. 1h, Supplementary Data 1). This variant leads to an in-frame deletion of a serine within the Zn-finger motif. Within the three MAC patients we report, no candidate pathogenic variants were found in the known MAC genes present in the current Genomics England PanelApp (ocular coloboma v1.47, anophthalmia or microphthalmia v1.51, structural eye disease v3.79). Thus, these cases further support that rare variants in *NR6A1* can cause MAC with reduced penetrance.

### Molecular Modeling Suggests Missense Variants Disrupt Important Intramolecular Interactions

The NR6A1 amino acid sequence is well-conserved between human, mouse, and zebrafish; specifically, the residues Ser76, Arg92, and Arg436 are conserved across multiple species (Supplementary Fig. 3). To understand the effects, missense variants had on protein stability and function, we created an in-silico model of a complex of NR6A1 with DNA (Supplementary Fig. 4a). The AlphaFold model of NR6A1 is shown by the composition of Zn-finger (residues 60–172) and NR_LBD (residues 246–480) domains shown in orange and green, respectively. The rest of the model shown in gray is predicted as an irregular structure by AlphaFold. In wild-type (WT) NR6A1, a positively charged arginine residue 92 is predicted to interact with negatively charged DNA based

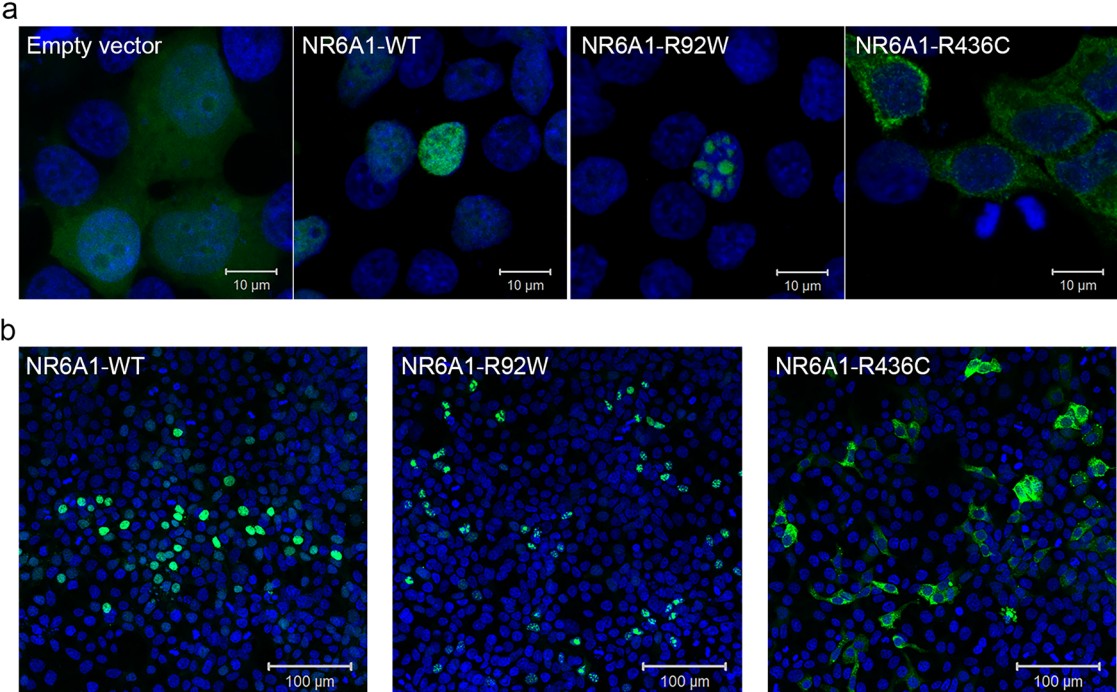

**Fig. 2 | Subcellular localization of wild-type (WT) and mutant forms of NR6A1.** NR6A1 variant localization pattern was studied by overexpression in HEK293 cells and representative high magnification (63X) images are shown from three different trials (**a**) Scale bar = 10 μm. Low magnification images (**b**) scale bar 100 μm. The localization pattern for the WT and the two variant isoforms was observed to be consistent across three transfection experiments. (Cells counted: WT = 387, R92W = 350 and R436C = 217).

on a zinc-finger protein model (Supplementary Fig. 4b). The R92W variant replaces the R92 residue with hydrophobic tryptophan (W), possibly interrupting the electrostatic interaction with DNA. The R436C variant affects the putative nuclear receptor ligand binding domain NR_LBD. In NR6A1, hydrogen atom 1HH2 of arginine R436 closely interacts with the oxygen atom of glutamic acid E388 (Supplementary Fig. 4c). The variant R436C breaks this bond and creates a cysteine residue which could form abnormal disulfide bridges in the variant protein, since residues C443, C391, and C422 are distanced at 8–12 Å from C436 in this variant domain as compared to 14–19 Å (C443–C391), 11.09 Å (C443–C422) and 4.52 Å (C91–C422) in the WT protein model.

**Missense variants alter NR6A1 protein subcellular localization**
To study the functional impact of the missense variants on protein localization in the cell, the R92W and R436C mutations were introduced in WT *NR6A1* cDNA fused to a GFP coding sequence. All experiments were performed in context to the *NR6A1* isoform NM_033334.4 and repeated at least three times. Transfection efficiencies were between 50-60% for the WT and variant constructs as analyzed by flow-cytometry and Western blotting (Supplementary Figs. 5, 6). The WT-*NR6A1* when over-expressed in HEK293 cells was consistently observed to localize in the nucleus (Fig. 2a, b), consistent with a previous report[22]. The R92W variant, although nuclear, was not uniformly distributed across the nucleus. To study the localization of R92W variant puncta to the nucleolus, we performed immunofluorescence staining of NR6A1 wild-type and mutant isoforms with nucleolar marker FIBRILLARIN. As shown in supplementary Fig. 7, the punctae do not colocalize with FIBRILLARIN staining suggesting that the variant is not mis-localized to the nucleolus. In contrast, the R436C variant localized exclusively in the cytoplasm (Fig. 2a, b). The above-described localization pattern of the WT and variant isoforms was consistent in all transfected cells and across multiple rounds of transfection. Taken together these results suggest

that both missense variants likely interfere with NR6A1 function due to improper subcellular localization.

**Expression pattern of mouse and zebrafish NR6A1 homologs suggests a role in early eye, kidney, and somite development**
Analysis of bulk RNA-Seq datasets from ocular and non-ocular tissues demonstrates modest expression of *NR6A1* in most tissues and relatively higher levels of expression in embryonic stem cells/induced pluripotent stem cells (compared to adult ocular tissues) and in bone marrow and testis systemically (Fig. 3a, d)[23,24]. Consistent with this observation, bulk RNA-Seq data from human fetal tissue shows that NR6A1 expression is highest in early stages of development, including the time of optic fissure closure in the first trimester[25,26] (Fig. 3b). In the Human Retinal Cell Atlas single nucleus RNA-Seq dataset, *NR6A1* is highly expressed in adult horizontal cells and low in microglia and RPE (Supplementary Fig. 8)[27]. Expression of *NR6A1* is strongly correlated (>5 fold enrichment, p = 0.0024) with that of other coloboma-associated genes in fetal ocular tissues (Fig. 3c). This strength of enrichment was not seen in Genotype-Tissue Expression (GTEx) body tissue (p = 0.361) or adult eye tissue (p = 0.451)[23,24]. We note that several of the highly correlated genes-*SALL4* (Duane-Radial Ray Syndrome), *PAX2* (Papillorenal syndrome), *ACGT1* (Baraitser-Winter Syndrome 2), *SALL1* (Townes-Brocks Syndrome 1) can also present with congenital renal anomalies.

To establish plausible causation for *NR6A1* variants, we studied the embryonic expression of its orthologs in mouse and zebrafish model systems at developmentally relevant time points. Previous work has demonstrated widespread expression of *Nr6a1* in mouse at E8.5 and E9.5 (including the optic vesicle) that becomes nearly undetectable by E12.5[21]. To study expression in the optic cup around the time of optic fissure closure, we used a probe that detects all validated transcripts of mouse *Nr6a1* at embryonic day 10.5 (E10.5, early optic cup) and E11.5 (time of optic fissure closure). The manufacturer's control probe against a bacterial sequence was used for reference (Supplementary Fig. 9). At E10.5, we noted diffuse low-level expression

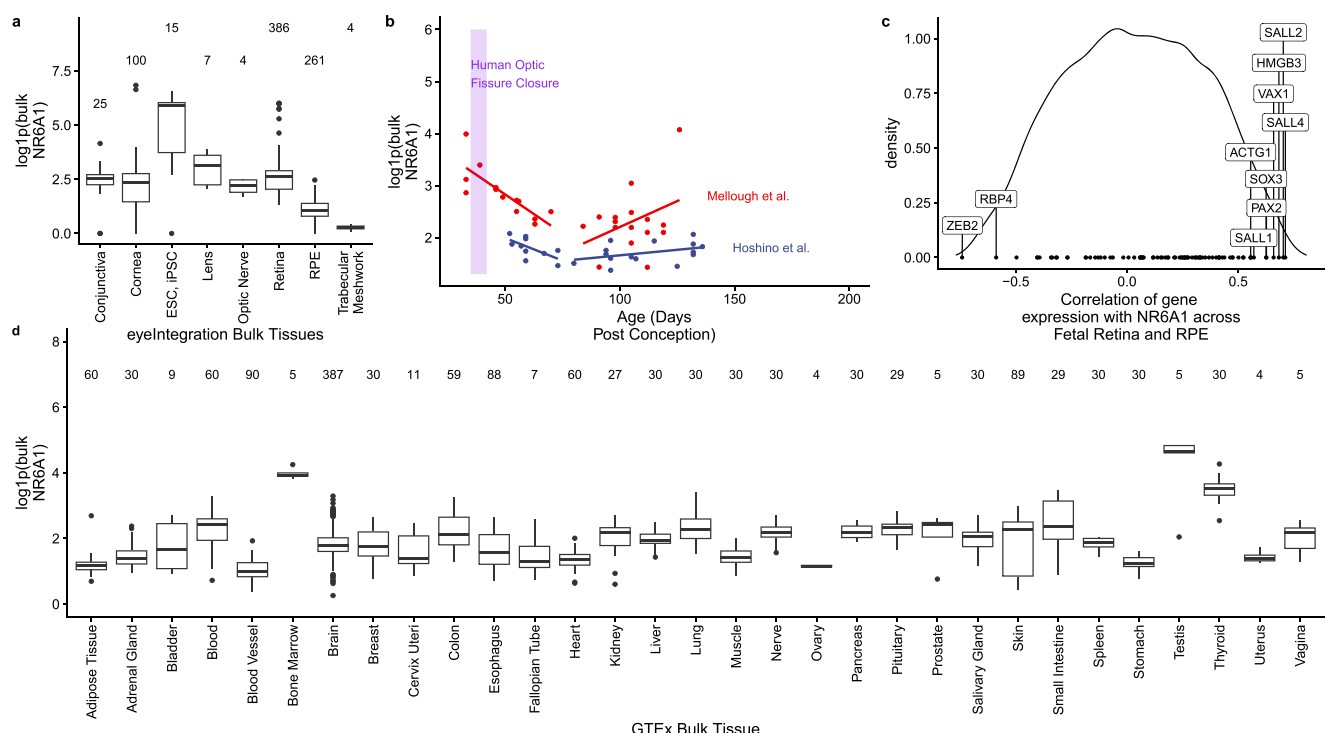

**Fig. 3 | NR6A1 gene expression in ocular and body tissues. a** Comparative levels of NR6A1 from publicly available bulk human tissue RNA-sequencing (RNA-Seq) datasets accessed on the eyeIntegration website (https://eyeintegration.nei.nih.gov/). On average, expression is higher in embryonic and induced pluripotent stem cells (ESC, iPSC, respectively) than in adult ocular tissues. **b** Bulk RNA-Seq data in human retinal fetal tissue from two studies suggests NR6A1 expression is highest in early stages of development, including the window of optic fissure closure (lavender box). NR6A1 expression is plotted against the tissue age (days post conception, dpc). A linear regression analysis was added for each paper's data from the 40 to 80 dpc and 80 to 160 dpc samples. **c** The density correlation plot (closer to −1 and 1 is more negatively or positively correlated, respectively) shows ten notable coloboma associated genes with highly ranked correlations with NR6A1 expression across eyeIntegration curated fetal retina and RPE tissues. This enrichment was not seen in adult tissues. **d** Among systemic tissues, NR6A1 is expressed most highly in bone marrow and testes. The boxplots display the median, 25th and 75th percentiles, and 1.5 * interquartile range (IQR). Any data outside the 1.5 * IQR are plotted. In panels a and d the number of samples is given above the boxplots.

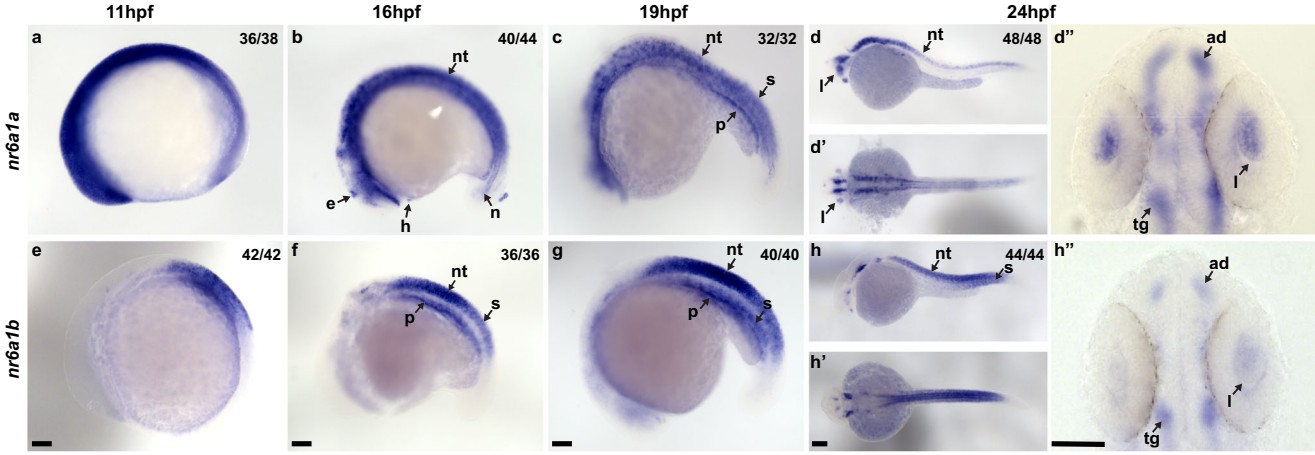

**Fig. 4 | Expression pattern of nr6a1a and nr6a1b paralogs in zebrafish.** nr6a1a is expressed ubiquitously at 11 hours post-fertilization (hpf) (**a**). By 16–19 hpf (**b, c**) expression is present in multiple structures including the somites (S), neural tube (NT) and notochord (N). At 24 hpf, expression remains in the NT but is decreased in the S and N. Expression in the lens (L) is first noted at 19 hpf and is particularly prominent by 24 hpf (**d–d'''**). nr6a1b expression at 11 hpf (**e**) is in anterior trunk, localizing to neural tube and somites from 16 hpf (**f**) and 19 hpf (**g**). At 24hpf (**h–h''**) it remains expressed in the neural tube and somites, with faint expression can be seen in the lens. All embryos are oriented in a lateral view, anterior to the left and dorsal up, except (**d', d'', h'**, and **h''**) shown in dorsal views. Scale bar = 100 μm. e-epiphysis, l-lens, p-pronephros, h- heart, n-notochord, s-somite, nt-neural tube, ad-anterior diencephalon, tg-tegmentum.

throughout the early optic cup and surrounding tissues that becomes significantly decreased by the time optic fissure closure commences (E11.5) (Supplementary Fig. 9). The level of expression in the optic cup is comparable to that observed in the brain vesicle, but higher than that observed in some areas of the surrounding eye mesenchyme.

In zebrafish, nr6a1 has two paralogs, nr6a1a and nr6a1b, both of which are maternally expressed[28,29]. At 11 hpf, when the optic vesicle evaginates, nr6a1a is widely expressed throughout the embryo, especially rostrally, showing less expression towards the posterior embryo axis (Fig. 4a, Supplementary Fig. 10a). Expression of nr6a1a is seen in

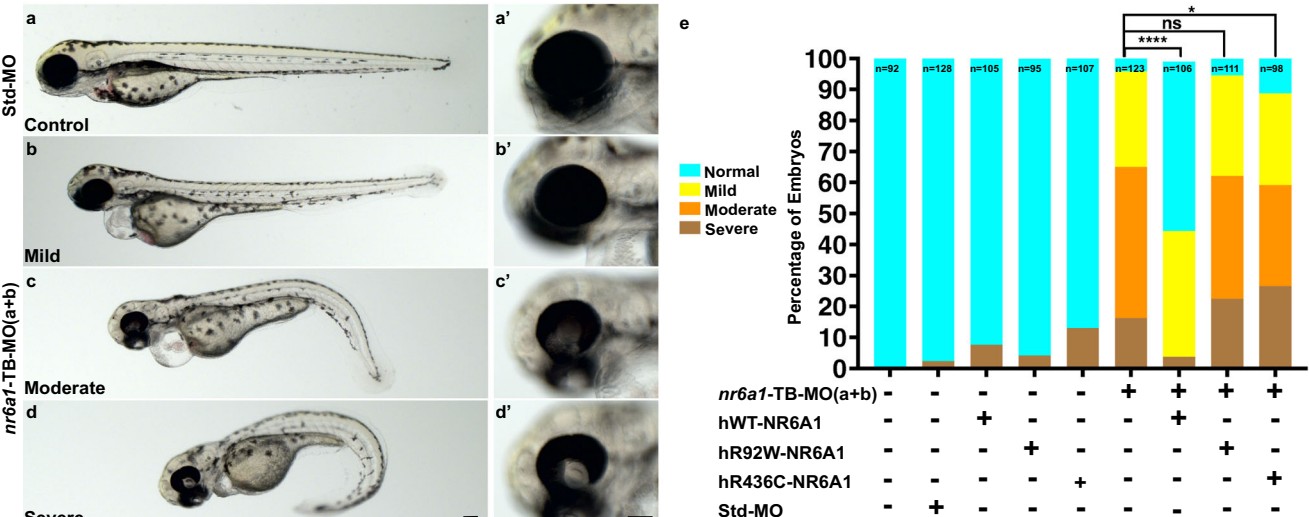

**Fig. 5 | Rescue of *nr6a1*+*nr6a1b* zebrafish morphant phenotypes with wild-type and mutant human NR6A1 mRNA.** Controls (**a**, **a'**) have a straight body axis and the optic fissure (OF) is closed. The *nr6a1*+*nr6a1b* morphants that have a mild phenotype (**b**, **b'**) have close to a normal body with microphthalmia and heart edema; a moderate phenotype (**c**, **c'**) with a slightly bent body axis with smaller eyes, coloboma and a severe heart edema; and severe morphants (**d**, **d'**) have a curved body axis with smaller eyes, coloboma and heart edema. The morphant phenotype was rescued when the morpholinos were co-injected along with the human-*NR6A1*-wild-type mRNA. However, there was no significant rescue in the morphant phenotype when the morpholinos were injected with either R92W or R436C human disease-causing variants (**e**). Morpholinos were injected at 0.75 ng each (1.5 ng total). Embryos were imaged at 72 hpf. Scale bar = 100 μm. Statistical significance was calculated using Chi-square test and Fisher's test. P value of *nr6a1*-TB-MO(a + b) V hWT-NR6A1 is <0.0001 and between *nr6a1*-TB-MO(a + b) V hR92W and hR436C are 0.0266 and 0.5169 respectively. Source data file has been provided for (**e**).

heart and periocular tissue at 14 hpf (Supplementary Fig. 11); expression in the heart reduced by 16 hpf and is absent by 19 hpf (Fig. 4b, c). At 16 hpf, *nr6a1a* remains widely expressed, while becoming restricted to the ventral regions of the brain, epiphysis, periocular tissues, heart and in the notochord and neural tube (Fig. 4b); expression in the developing eye is reduced compared to the adjacent developing brain (Supplementary Fig. 10b). Notably, *nr6a1a* expression is absent from the neural-mesodermal progenitor region in the tail of zebrafish embryos, consistent with its role in the trunk differentiation program[21]. By 19 hpf the expression appears to decrease overall but remains present in the ventral brain regions, notochord, somites, and the pronephric duct (Fig. 4c). At 24 hpf, expression is prominent in the anterior diencephalon, tegmentum, midbrain, and along most of the length of the embryo in the neural tube; interestingly, expression is nearly absent from the neural retina and retina pigmented epithelia but is prominent in the lens (Fig. 4d–d"). After 26 hpf and up to 72 hpf we observed no detectable *nr6a1a* expression, consistent with published single-cell mRNA expression during zebrafish development[28,29].

Unlike *nr6a1a*, *nr6a1b* expression at 11 hpf is limited to a patch in the posterior neuroectoderm of the embryo but excluded from the most caudal region (Fig. 4e). At 16 hpf and 19 hpf, *nr6a1b* expression is prominent in the neural tube, somites, and pronephric duct and, like *nr6a1a*, is excluded from the neural-mesodermal progenitor region in the tail (Fig. 4f, g). By 24 hpf, expression is decreased in most tissues but remains in the tegmentum, cranial ganglia, neural tube, and somites in the distal region of the trunk (Fig. 4h-h"). By 36 hpf and through 72 hpf, *nr6a1b* is notably expressed in the developing lens, brain, and cranial ganglions. (Supplementary Fig. 12). At 72 hp we also note faint expression in the retina and in the presumed RPE (Supplementary Fig. 12d, e).

**Morpholino (MO) knockdown of zebrafish nr6a1a/nr6a1b recapitulates human phenotypes which are not rescued by pathogenic variant mRNA**

All MO experiments were carried out following the guidelines set forth for their use in zebrafish[30–32]. These guidelines include: 1) use of two non-overlapping MOs (one translation blocking (TB), one splice blocking (SB)); 2) observation of a consistent phenotype with both TB and SB MOs for each paralog; 3) a correlation between dose of MO and phenotype, with lower concentrations of MO causing a milder phenotype; 4) validation of the efficacy of SB MOs by RT-PCR analysis; 5) lack of a phenotype with injection of a control MO; and; 6) partial rescue of the MO phenotype with co-injection of the corresponding human mRNA.

To test the functional consequences of *nr6a1a* and *nr6a1b* knockdown, we designed TB and SB MOs for each paralog of the gene. The sequence of the TB morpholinos does not overlap with the human mRNA sequence and is therefore unlikely to interfere with mRNA rescue experiments (Supplementary Fig. 13). Morphants were divided into four phenotypes: normal, mild (normal/near normal body axis with microphthalmia), moderate (slightly shortened and mildly curved body axis, microphthalmia ± coloboma and heart edema) or severe (significantly shortened and curved body axis, microphthalmia ± coloboma, heart edema) (Fig. 5a–d', Supplementary Figs. 14, 15). Embryos were scored at 72 hpf (after optic fissure closure and initial stages of eye growth are normally completed) to ensure microphthalmia/coloboma represents a true phenotype and not because of developmental delay or undergoing growth compensation.

Knockdown of *nr6a1a* (Supplementary Fig. 14e) or *nr6a1b* (Supplementary Fig. 15e) with either TB-MO or SB-MO resulted in a significant number of moderate and severe phenotypes with few mild phenotypes. Although the effect of TB-MO and SB-MO were similarly potent for *nr6a1b* knockdown, the SB-MO had a stronger effect than the TB-MO for *nr6a1a*. SB-MO knockdown of the gene was validated for both paralogs by reverse transcription-PCR experiments (Supplementary Figs. 14f, 15f). MOs can elicit p53-activated cell death and cause non-specific defects[33]. We therefore tested MOs for both *nr6a1* paralogues at all concentrations used with and without co-injection of a p53-MO. The phenotypic spectrum was not affected by co-injection of this p53 MO, suggesting widespread cell death was not the primary cause of our observations (Supplementary Fig. 16).

Overexpression of 100 pg of human *NR6A1* mRNA in zebrafish shows no overt phenotype (Supplementary Fig. 17a, b). Co-injection

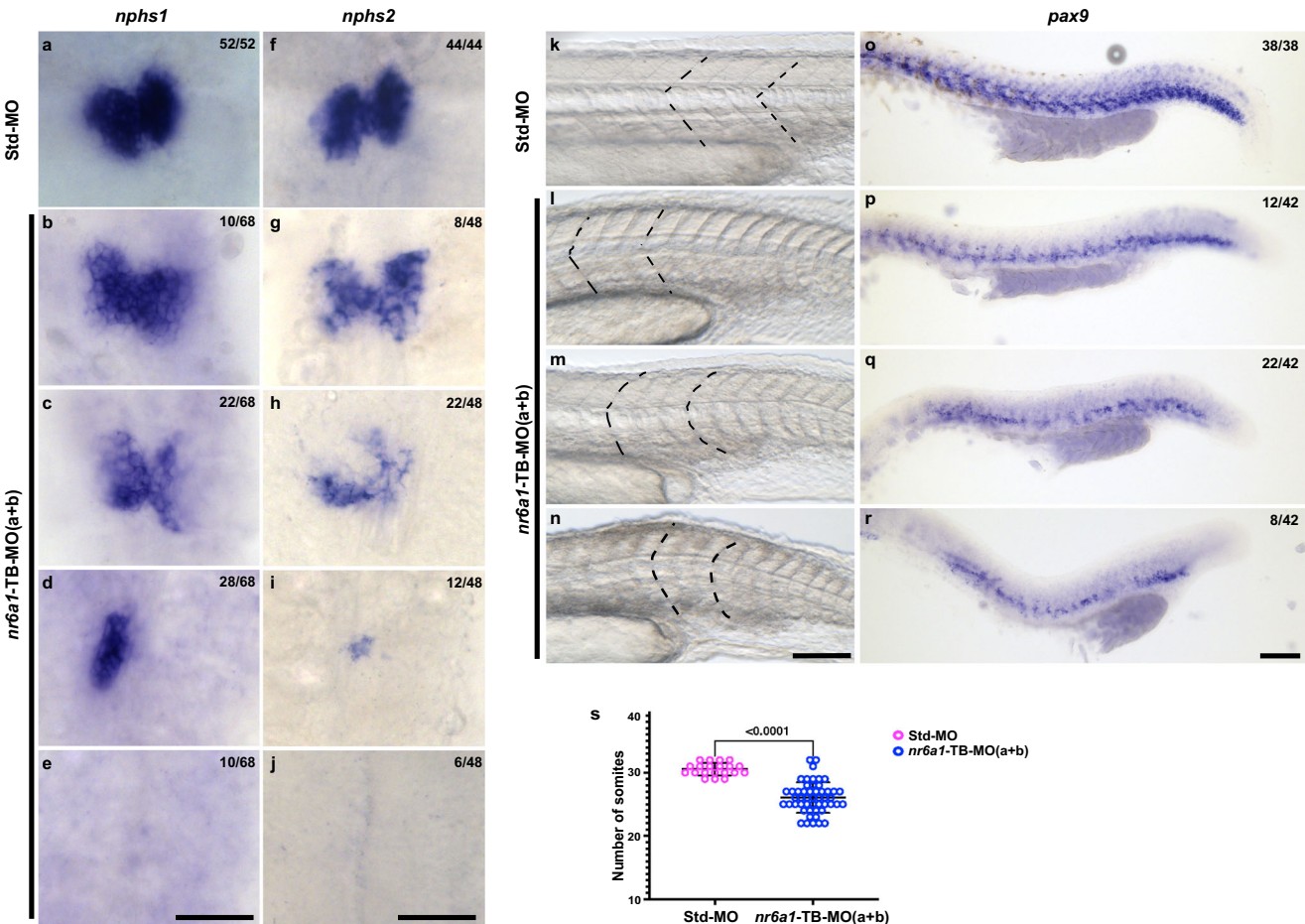

**Fig. 6 | *nr6a1(a + b)* zebrafish morphants have kidney and vertebrae defects.** At 48 hpf, control embryos demonstrate bilateral expression of *nphs1* (*nephrin*) and *nphs2* (*podocin*) (**a**, **f**), markers of kidneys. Knockdown of *nr6a1(a + b)* resulted in a range of abnormal expression patterns including mildly reduced expression (**b**, **g**), moderately reduced expression (**c**, **h**), unilateral or midline expression (**d**, **i**) or absent expression (**e**, **j**) of *nphs1* and *nphs2*. Somites (the early precursors of vertebrae) have a sharp, chevron shape with an approximately 90-degree angle in control zebrafish at 24 hpf (**k**). Knockdown of *nr6a1a* and *nr6a1b* resulted in a range of abnormal expression patterns including chevron shapes with obtuse angles (**l**), as well as mildly rounded (**m**) or flattened (**n**) somites. The sclerotome marker *pax9* is expressed in a continuous angular pattern in the ventromedial portion of somites along the length of 24 hpf control embryos (**o**). Double morphant embryos exhibit a spectrum of abnormal patterns of *pax9* expression including reduced levels of expression (**p**), patchy expression extending beyond the caudal tip of the yolk (**q**) and patchy expression ending near the caudal end of the yolk sac (**r**). *nr6a1(a + b)* morphants have reduced number of somites (**s**), statistical significance is calculated using two tailed t-test. Std-MO n = 20 and *nr6a1*-TB-MO(a + b) n = 50. Data is represented as difference between means of *nr6a1*-TB(a + b)−Std-MO ± SEM (−4.490 ± 0.5629). P value is <0.0001. 0.75 ng of each paralogue is used for knockdown. Numbers of each representative pattern are given in each panel. Scale bar = 100 μm. Source data file has been provided for (**s**).

of 2 ng and 1.25 ng of nr6a1a and nr6a1b, TB-MO respectively along with 100 pg of WT human mRNA (hWT-*NR6A1*), resulted in a rescue, with over 60% embryos exhibiting a normal/control-injected phenotype, thus validating our TB-MO's (Supplementary Figs. 14g, 15g). In contrast, co-injection with either hR92W or the hR436C missense variants of *NR6A1* identified in coloboma patients were significantly less effective in rescuing the zebrafish *nr6a1a/b* knockdown, indicating that the missense variants are deleterious (Supplementary Figs. 14g, 15g).

To study the effect of knocking down of both *nr6a1a* and *nr6a1b* zebrafish paralogues, we co-injected 0.75 ng of TB-MO for each paralog (1.5 ng total), resulting in a similar spectrum of eye and body axis phenotypes compared to the knockdown of individual paralogues (Fig. 5a–d, a′–d′). At 48 hpf, double morphants exhibited abnormal expression of *nphs1/nephrin* and *nphs2/podocin*, both of which are required for the formation, maturation and maintenance of kidneys[34–36] (Fig. 6a–j). The spectrum of expression patterns included reduced, absent, midline and asymmetric expression of *nphs1/2* (Fig. 6d, h) which persisted at 72 hpf (Supplementary Fig. 18).

Vertebrae develop from the sclerotome of somites during development[37,38]. At 24 hpf, control embryos have chevron shaped somites, while morphants exhibit a spectrum of abnormal morphologies ranging from blunting of the chevron angle to more severe U-shaped or flattened somites (Fig. 6k-n). Double morphants also exhibit a significantly decreased number of somites, consistent with the missing vertebrae phenotype in several patients (Fig. 1, Fig. 6s). Similarly, the sclerotome marker *pax9*[39,40] is expressed in a uniform and regular pattern in the ventromedial region of somites of control embryos at 24 hpf (Fig. 6o). By comparison, double morphants show varying degrees of decreased and/or patchy *pax9* expression (Fig. 6p-r). Overall, our results indicate that it is likely that *nr6a1a + b* morphant embryos have defective vertebrae development.

Injection of TB-MO's, *nr6a1(a + b)*, resulted in 16% and 49% embryos having severe and moderate phenotypes respectively; co-injection of 100 pg hWT-*NR6A1* mRNA, resulted in >50% embryos having straight body and normal eye, however 19% of these embryos show, heart edema. Breaking down each phenotype separately, we rescued approximately 55% embryos for coloboma, 53% for body axis

and 44% for heart edema. Neither the hR92W nor hR436C *NR6A1* mRNAs resulted in significant rescue, of any of the described phenotypes, confirming the pathogenicity of these variants (Fig. 5e, Supplementary Data 5). Injection of 0.75 ng of either *nr6a1a* TB-MO or *nr6a1b* TB-MO resulted in a significantly milder phenotype, suggesting that co-injection of these had at least an additive phenotypic effect in the combined MO injection experiment (Supplementary Fig. 19).

A prior study reported that both overexpression and loss-of-function of *nr6a1* can result in developmental phenotypes in *Xenopus laevis*[41], we also evaluated the effect of injection of human *NR6A1* mRNA on zebrafish development. Overexpression of 150 pg of *hNR6A1* mRNA resulted in microphthalmia and heart edema with a straight body axis (n = 91/108) (Supplementary Fig. 17c, c'). At 200 pg, overexpression of *hNR6A1* mRNA, phenotypes were more severe including colobomatous microphthalmia, heart edema and a bent body axis (n = 60/92), with 26% (n = 24/92) (Supplementary Fig. 17 d-f'), exhibiting noticeable shortening and loss of chevron-shaped somites (Supplementary Fig. 20); a minority of embryos (n = 8/92) developed no discernible eyes (Supplementary Fig. 17f, f'). Taken together, these experiments demonstrate that normal zebrafish eye development is sensitive to *nr6a1* dosage and both reduced and increased *nr6a1* expression result in developmental phenotypes analogous to human colobomatous microphthalmia.

## Discussion

Here we describe six *NR6A1* variants that cause an autosomal dominant syndromic form of colobomatous microphthalmia and missing vertebrae with or without congenital kidney abnormalities, which we term OVR syndrome. As with many other cases of syndromic and non-syndromic microphthalmia/coloboma, the OVR syndrome show, incomplete penetrance and variable expressivity[1]. By 2015 ACMG/AMP variant interpretation criteria, we considered chr9:g.124536516_124643457del pathogenic (criteria: PVS1, PP1_Strong, PM2) and other MAC-associated variants likely pathogenic (criteria: Ser76del, PM1, PM2, PM4, PP3; Arg92Trp, PS3, PM1, PM2, PP3; Arg436Cys, PS3, PM2, PP3; Ser301Ter & Ser322Ter, PVS1, PM2). Thus, *NR6A1* variants were causative among 1.3%–1.4% families in two independent patient cohorts (3 out of 224 in the NEI coloboma cohort and 3 out of 215 in the MAC cohort in the UK100KGP).

The NEI study, which specifically recruits patients with coloboma/microphthalmia, performs extensive phenotypic analysis on probands including complete eye examination, kidney ultrasound, neuropsychological testing, physical exam/dysmorphology exam, spine x-ray, routine bloodwork/urinalysis, ECHO (in the presence of a murmur), and audiology. Additional testing (e.g., brain MRI) may be performed on an as needed basis. In addition, all available first-degree relatives undergo a complete dilated fundus exam. As such, we have greater certainty that a patient is truly unaffected, say, by coloboma, rather than being simply asymptomatic. Indeed, the mother of the proband in family COL034 (COL034.2), for example, was visually asymptomatic and unaware of a *forme fruste* of coloboma or a missing thoracic vertebra prior to her exam with us. Conversely, the Genomics England database spans an entire population in a gene and phenotype-agnostic manner but may contain incomplete or unrelated phenotypic information. As such, phenotypes such as intellectual disability (Individual B1, Supplementary Data 1) may be spurious associations or may be uncommon manifestations of an *NR6A1*-related syndrome. Confirmation of these and other possible phenotypes awaits description of additional cases. We include congenital renal disease as part of this syndrome not only because two individuals in two separate pedigrees exhibited these phenotypes, but also because Rasouly et al. have simultaneously identified presumed loss-of-function variants in thirteen individuals with congenital renal abnormalities, with or without congenital eye abnormalities, providing further validation of our findings[42]. Using a combination of imaging and genetic data, Sun et al.

recently reported that NR6A1 was a key gene associated with differences in vertebral number[43]. In addition, Jacquinet et al. subsequently noted congenital kidney, uterine, and vertebral anomalies in three patients and in zebrafish mutant line[44]. None of our patients reported uterine abnormalities or difficulty with pregnancy; however, pelvic ultrasounds were not generally performed, so subclinical phenotypes cannot be ruled out. Because none of our affected patients displayed heart defects, we currently do not include this as part of the syndrome, even though heart edema was noted in the morphants. However, we cannot rule out that congenital heart anomalies will be found in subsequent patients as more are identified.

Additional studies are required to understand the detailed disease mechanisms of *NR6A1* variants. Deletion and presumed truncating variants are generally associated with a haploinsufficiency mechanism such as nonsense-mediated decay. This may be tested by expression profiling in patient cells. However, the subcellular localization defects of the two missense variants in *NR6A1* hint at more than one mechanism of disease. The early expression of *NR6A1* homologs in mouse and zebrafish are consistent with the previous data[21] and suggest that the colobomatous microphthalmia observed in our patients may result from effects on early eye morphogenesis rather than a defect in optic fissure closure per se. Given the known roles of *Nr6a1* in stem cell biology, we posit that the developmental trajectory of the optic cup neuroepithelium is altered in a way not consistent with optic fissure closure. However, given the expression of *nr6a1a/nr6a1b* in the lens vesicle in zebrafish and mouse, a non-cell autonomous effect on optic fissure closure cannot be excluded. In fact, evidence from Mexican surface and cave fish (*Astyanax mexicanus*) experiments show that early neural retina development and maintenance relies on a healthy lens[45,46].

Recently, *NR6A1* has been shown to be important for somite development and, consequently, vertebral number, thus strengthening the phenotyping link with missing vertebrae we describe in humans[18–21,47]. Vertebrae differentiate from somites which develop their stereotyped segmentation pattern in an anterior to posterior progression during early development, with successive *HOX* genes specifying different regions of the spine via a process called temporal collinearity. Homozygous germline inactivation of *Nr6a1* in mice results in embryonic lethality around E10.5 with cardiovascular, neural tube and hindgut abnormalities as well as fewer somites (13, rather than the normal 25)[21,48]. In S*us domesticus* (pig), *NR6A1* was identified as a quantitative trait locus for vertebral number, which is known to vary between breeds[18,20]. In *Equus assinus* (donkey), an *NR6A1* intronic polymorphism is associated with body size/vertebral number and a single nucleotide polymorphism in exon 8 is associated with the number of lumbar vertebrae in Kazakh sheep[19,47]. In developing *Xenopus*, *NR6A1* is expressed in late tailbud and neurula stages; overexpression results in posterior defects and disturbed somite formation, while expression of a dominant negative form of the receptor results in abnormal neural tube differentiation, loss of head structure including eyes[41], and downregulation of a retinoic acid receptor (RARγ2) anteriorly[49]. Retinoic acid treatment of embryos upregulates expression of NR6A1, increasing primary neurogenesis via factors such as NeuroD, XDelta1 and x-ngnrl. Retinoic acid is a known and important regulator of both ocular and kidney development[50,51]; whether retinoic acid receptor signaling is disrupted in model systems of *Nr6a1/nr6a1* is currently under investigation. However, all phenotypes previously observed when modulating the activity of *NR6A1* in animal models are consistent with the developmental defects in the eyes, kidneys, and vertebrae that we observe in patients carrying deleterious mutations in NR6A1.

In this study, we identified novel *NR6A1* variants in three unrelated families with an OVR syndrome; these findings were further corroborated in an independent cohort using a genome first approach. Using in silico prediction and molecular studies we demonstrated that these

highly conserved variants disrupt NR6A1 protein structure leading to mis-localization at the cellular level. We further demonstrated enrichment of coloboma-associated genes with *NR6A1* in fetal, but not adult tissues. Expression of *NR6A1* homologs in mouse and zebrafish embryos suggests disease relevant tissue-specific gene expression pattern. This was further confirmed by in vivo experiments where the knockdown of zebrafish *nr6a1a and nr6a1b* resulted in ocular, renal and vertebral phenotypes that were partially rescued with WT human *NR6A1* mRNA but not with the two variants tested. This data implicates the human *NR6A1* gene variants with the OVR syndrome.

## Methods

### Patients and clinical studies
The design and the conduct of the study complied with all relevant regulations governing research on human subjects and according to the principles of the Declaration of Helsinki. Complete eye examinations and genetic testing at the National Eye Institute (NEI) were conducted after informed consent from all participants under National Institutes of Health-Internal Review Board-approved clinical protocols (NCT01778543, NCT01087320, NCT02077894, www.clinicaltrials.gov). This consent included permission to publish deidentified data. No compensation was provided to the participants. Probands underwent systemic testing as clinically indicated, which included physical exam, kidney ultrasound, routine blood chemistries, audiology, and spine x-ray. Eye examinations included age-appropriate testing of visual acuity, refraction, ocular motility/alignment, slit lamp exam, dilated fundus exam, and ophthalmic photography. Specific informed consent for exome/genome sequencing was obtained under an IRB-approved protocol, along with pre- and post-test genetic counseling (NCT02077894). Family COL005 and COL034 were previously reported as Family 1 and 2, respectively, without molecular characterization and detailed individual phenotyping data[16]. For patients and relatives recruited from the Genomics England 100,000 Genomes Project (UK100KGP), informed consent for whole genome sequencing (GS) was obtained in accordance with approval from the HRA committee East of England-Cambridge South (REC 14/EE/1112)[17]. Sex was not considered as an independent variable as OVR syndrome is autosomal dominant, affecting both males and females, and due to the relatively small size of the studied families. Sex reported in Table 1 is self-identified by the participants.

### Genetic testing
Genomic DNA samples prepared from blood or saliva from NEI patients, and their family members were subjected to short-read Next-generation sequencing (NGS) using Illumina platforms. In total, 101 proband samples were subjected to amplicon sequencing of the *NR6A1* gene (Supplementary Data 2) using a MiSeq sequencer (2 × 300 bp paired-end), 57 samples subjected to exome sequencing (2 × 150 bp paired-end, xGen exome v1 supplemented with additional probes, Blueprint Genetics), 66 samples subjected to GS (2 × 150 bp paired-end, PCR-free library, NIH Intramural Sequencing Center). Reads were aligned to the GRCh38 reference genome, small variants and structural variants were then called, annotated, and prioritized using a custom NGS analysis pipeline (https://github.com/NIH-NEI/NGS_genotype_calling & https://github.com/NIH-NEI/variant_prioritization).

Sanger sequencing was performed to confirm select variants in probands and family members using the BigDye-direct sequencing kit (Thermo Fisher) using primers provided in Supplementary Data 2. The deletion breakpoint in family COL005 was also determined by PCR and Sanger sequencing (Supplementary Data 2). Breakpoint PCR was further used for genotyping of the COL005 family. The logarithm of the odds (LOD) score in family COL005 was estimated using the formula $\log_{10}(1/0.5^{\text{Segregations}})$.

Additional patients and family members underwent Genome Sequencing (GS) as part of the UK100KGP including the clinical variant

interpretation pipeline (The National Genomics Research Library v5.1, Genomics England. doi:10.6084/m9.figshare.4530893/7. 2020.). Genome data from affected individuals recruited with a clinical phenotype in keeping with microphthalmia, anophthalmia or coloboma were interrogated for rare (minor allele frequency <0.001, gnomAD v3.1 dataset) biallelic or de novo protein altering variants across the genome. Candidate variants underwent manual curation including in silico prediction, literature search and pathway analysis to establish biological plausibility as a pathogenic variant in developmental eye disease. Additional analyses of all rare protein altering variants in *NR6A1* across the entire UK100KGP was performed to identify any individuals outside of the ophthalmology cohort who harbored a candidate pathogenic variant. All variants were manually inspected in the Integrative Genomics Viewer (IGV) after loading sample bam files. Variants appeared to be artifacts were not reported.

### Variant classification
The 2015 ACMG/AMP sequence variant interpretation guidelines were followed for variant classification[52,53]. The PM1 (functional domain) criterion was applied to variants in part of the DNA binding domain, a.a. Thr68-Lys119, as the region is highly constrained for missense variations in gnomAD (v2.1.1, missense observed/expected = 0.19, p value = $6 \times 10^{-6}$). The PP3 criterion was applied to missense variants based on a collection of in-house in silico prediction tools (https://github.com/NIH-NEI/variant_prioritization) and the inframe deletion variant based on five in silico prediction tools (CAPICE, FATHMM-indel, MutationTaster, MutPred-Indel, and SIFT).

### Molecular modeling
A structural model of NR6A1 was generated using the AlphaFold server, AF-Q15405-F1-model_v4). The Zn-finger domain (ZFD) and nuclear receptor ligand binding domain (NR_LBD) were saved as two PDB files. The binding of DNA to the ZFD of NR6A1 was modeled using a single ZFD domain of the retinoid X receptor alpha-liver X receptor beta (PDB ID: 4NQA) in a complex with DNA. Two variants (R92W and R436C) were generated using the Edit > Swap > Residue function on the respective domain PDB files in YASARA. Variant models were optimized and minimized using gradient descent. All two minimized mutants and the two WT, ZFD and NR_LBD models were subjected to 10 ns of Molecular Dynamics (MD) using YASARA's "run.mcr" macro. Ion concentration was added as a mass fraction with 0.9% NaCl. The simulation temperature was set to 310 K with a water density of 0.997 g/mL. For each domain, the cell size extended to 10 Å beyond each side of the protein in the shape of a cube. Dimensions were 90.2 Å × 90.2 Å × 90.2 Å and 82.5 Å × 82.6 Å × 82.6 Å for the nuclear receptor Zn-finger and ligand-binding domains, respectively. Each simulation was run in YASARA using an AMBER14 forcefield, with a timestep of 2.5 fs. Simulation snapshots were outputted for every 0.1 ns, resulting in 100 simfiles for each simulation.

### Fish maintenance and zebrafish strains
*Danio rerio* were maintained under standard conditions. Embryos were staged according to Kimmel et al., 1995[54]. ABTL stocks were used for all the experiments, which were carried out in accordance with National Eye Institute, Animal Care and Use Committee Protocol Number NEI-648 and NIH Animal Research Advisory Committee.

### Zebrafish in situ hybridization
Embryo were fixed in 4% paraformaldehyde (PFA) overnight at 4 °C and dehydrated in methanol for 1 h at −30 °C. The embryos were rehydrated, treated with proteinase-K and re-fixed with 4% PFA. Pre-hybridization and hybridization were carried out at 65 °C. RNA probes were synthesized using a DIG labeling kit (Millipore-Sigma, 112770739) following manufacturer's protocol. *nr6a1a* RNA probe was synthesized from a CDS clone in TOPO TA vector (ThermoFischer Scientific), while

*nr6a1b* was synthesized using PCR product as a template. Primers are noted in Supplementary Data 3. Samples were hybridized overnight with RNA probes at 65 °C, washed, incubated with Anti-DIG antibody (Millipore-Sigma, 1109327490); color was developed using BCIP/NBT substrate (Millipore-Sigma, 11681451001) in alkaline phosphatase buffer. Embryos were imaged with Leica DM6 dissecting microscope.

## Morpholino gene knockdown and rescue experiments in zebrafish

All morpholinos (MO) were obtained from Gene Tools LLC. MOs used to target zebrafish *nr6a1a* and *nr6a1b* are given in Supplementary Data 4. Human *NR6A1*-wild-type, variants *NR6A1*-R92W and *NR6A1*-R436C DNA fragments were synthesized and cloned in pCS2+ (Azenta Life Sciences). Plasmids were linearized with *Not I* restriction enzyme and capped mRNA was synthesized using mMessage mMachine T7 Transcription kit (ThermoFischer Scientific). MOs and mRNA were co-injected into zebrafish embryos at single cell stage. *nr6a1a* and *nr6a1b* translation blocking (TB) MOs were used at 2 ng and 1.25 ng respectively. *Nr6a1a* and *nr6a1b*, SB-MOs were injected at 2 ng and 1 ng respectively. Human *NR6A1*-wild-type was used at 100 pg and 150–200 pg for RNA rescue and over expression studies respectively. *NR6A1*-R92W and *NR6A1*-R436C RNAs were used at 100 pg for rescue experiments. For over-expression experiments, doses of 100 pg–200 pg *hNR6A1* mRNA were injected at the single cell stage. Embryo phenotypes were scored and imaged at 72 hours post-fertilization (hpf) using Leica DM6 dissecting microscope.

## Cell culture and transfection studies

HEK293T cells maintained in DMEM with 10% FBS and 1% penicillin-streptomycin were seeded onto 4-well chamber slides, maintained for 24 h and transiently transfected with GFP tagged WT and/or mutant *NR6A1* constructs (Azenta Life Science, Burlington, MA, USA) using X-treme Gene HP (Roche, Indianapolis, IN, USA) following manufacturers' instructions. After 24–48 h of transfection, transfected cells were fixed for 15 min in 4% paraformaldehyde (PFA) in PBS. After washing with 1 × PBS cells were incubated for 1 h at room temperature with Hoechst33342 (1:250 dilution in PBST). Subsequently, the slides were washed and mounted with Fluoromount-G® (SouthernBiotech, Birmingham, AL, USA). Zeiss confocal microscopes 880 coupled with an Airyscan® detector was used for confocal imaging. The images were analyzed using ZEN Software (Carl Zeiss Microscopy LLC, Thornwood, NY). The cell culture experiments were repeated at least three times for each for variant localization studies. Co-transfected cells with both WT and mutant forms of the NR6A1 constructs were fixed for 15 min in 4% paraformaldehyde (PFA) in PBS. After washing with 1 × PBS and per-meabilization and blocking in ICC buffer (0.5% BSA 0.5% Tween and 0.1% triton X100 1 × PBS). Cells were then incubated overnight at 4 °C with the Fibrillarin antibody (MA3–16771, Thermo Fisher Scientific) in ICC buffer. After multiple washes in PBS, the cells were incubated for 1 hr at room temperature in Alexa Fluor™ 555 conjugated goat anti-mouse antibody (A-21422, Thermo Fisher Scientific) and Hoechst33342 (1:1000 dilution in ICC buffer). Cells were then washed in PBST before mounting with Fluoromount-G® (SouthernBiotech, Birmingham, AL, USA) imaging.

## Flow cytometry

Transfection efficiency was determined by measuring the expression of GFP after 48 hrs post transfection. HEK293 cells were detached from the plates using Trypsin for 5 min followed by neutralization with serum containing media. The cells were then fixed for 15 min in 4% paraformaldehyde (PFA) in PBS and then collected in 1× PBS containing 2% FBS (FACS buffer) and washed 2 times by centrifugation. The cell suspension was filtered through a 50 μm cell strainer. Data was acquired with a CytoFlex NUV instrument (Beckman Coulter, Brea CA) using the blue light excitation and 525 nm emission to detect GFP and violet light excitation and 450 nm emission to detect DAPI detection. Data analysis

was done using CytExpert software Version 2.5 (Beckman Coulter, Brea CA). Interesting cells were identified as DAPI negative, in the whole cell cluster in a FSC vs. SSC plot and being in a single cell state in the FSC-A vs. FSC-Width. Transfection efficiency was quantified as the Stain Index of GFP fluorescence intensity, which was calculated using the median fluorescent intensity and robust Standard Deviation as described. The cell culture experiments were repeated at least three times for each for variant localization studies.

## Mouse embryo in situ hybridization

C57Bl/6 mice (The Jackson laboratory, Strain #:000664) were housed in individually ventilated cages (five per cage) under conditions of a 14 h/10 h light/dark cycle and ambient temperature of 22 ± 2 °C with 30–70% humidity. Experiments were carried out in accordance with National Eye Institute, Animal Care and Use Committee Protocol Number NEI-605 and NIH Animal Research Advisory Committee. *Nr6a1* mRNA expression in mouse was assayed by RNA in situ hybridization with *Nr6a1*(Cat: 1314941-C1) probe using the RNAScope Assay, Multiplex fluorescent Reagent Kit V2 (Advanced Cell Diagnostics (ACD), Newark, CA, USA) on E10.5 and E11.5 cryosections[55]. The manufacturer's control against a bacterial sequence was used for comparison, RNAscope™ 3-plex Negative Control Probe (Cat. #320871). Exposure settings were identical for all samples to facilitate comparisons in expression levels. Expression levels in the eye were quantitated relative to brain and periocular tissue using a free open source bioimaging QuPath software[56] for comparison. A "best fit" model was applied to all samples analyzed.

## Antibody used in the study

FIBRILLARIN antibody from Invitrogen with Cat#-MA1-91878, Lot#-ZA389553, Clone#- DYKDDDDK Tag Monoclonal Antibody (FG4R) raised in mouse was used at 1:1000 dilution.

## Gene expression analysis of *NR6A1*

The h5ad (d27a79a1-8a5f-404d-8063-52e19122ef49.h5ad for adult and 88444d73-7f55-4a62-bcfe-e929878c6c78.h5ad for fetal) from the HRCA project were downloaded from cellxgene.cziscience.com and the raw counts were summed at the sample and cell type level to create a pseudobulk matrix with the python package ADPBulk (https://github.com/noamteyssier/adpbulk). The eyeIntegration (which includes GTEx) gene counts and sample level metadata were downloaded from eyeIntegration.nei.nih.gov (https://hpc.nih.gov/~mcgaugheyd/eyeIntegration/2023/gene_counts.csv.gz and https://hpc.nih.gov/~mcgaugheyd/eyeIntegration/2023/eyeIntegration23_meta_2023_09_01.built.csv.gz). The pseudobulk and bulk RNA-seq counts were normalized by counts per million (CPM) and transformed in R/4.3 to have a mean of zero and a standard deviation of one. Plots were created in R/4.3 with the ggplot2, cowplot, and ggbeeswarm packages.

## Correlation analysis of NR6A1

The same data used in the plotting was used to create separate gene expression matrices that for either fetal primary tissue (retina and RPE) or adult tissue (retina and RPE). We used spatial quantile normalization (spqn) analysis to derive gene correlation matrices that were not biased by higher gene expression. The same data used in the plotting was used to create separate gene expression matrices that for either fetal primary tissue (retina and RPE) or adult tissue (retina and RPE). We used spatial quantile normalization (spqn) analysis to derive gene correlation matrices that were not biased by higher gene expression[57]. Briefly, the first four principal components were removed with the WGCNA tool removePrincipalComponents. The correlation matrices were created with the base R "cor" function. The correlation scores were expression transformed with the spqn package's "normalize_correlation" function with the parameters "ngrp = 20, size_grp = 300, ref_grp = 18". The correlation matrix was filtered to only contain

correlations between NR6A1 and all other genes. Code for all bioinformatic analyses have been deposited in 10.5281/zenodo.14.

## Reporting summary

Further information on research design is available in the Nature Portfolio Reporting Summary linked to this article.

## Data availability

The genome sequencing data from NEI participants generated in this study have been deposited in the dbGaP database under accession code phs003996.v1. The genome data are available under controlled access due to restriction in participant informed consent; access can be obtained by following dbGaP data access policy. The application process, time frame for review of requests will be according to the published governance structure: https://sharing.nih.gov/accessing-data/accessing-genomic-data/how-to-request-and-access-datasets-from-dbgap. Data for all bioinformatic analyses have been deposited in 10.5281/zenodo.14757568. All other data are provided in the main text or in the Supplementary Information. Source data files are provided with this research article. Source data are provided with this paper.

## Code availability

Code for scRNA-seq and Multiple Sequence Alignment can be found at GitHub (https://github.com/davemcg/nr6a1/releases/tag/1.3)[58]. NGS analysis code can be found at GitHub (https://github.com/NIH-NEI/variant_prioritization/releases/tag/v0.1)[59].

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

## Acknowledgements

We are grateful to the patients and their families for participation in this research, which spanned over 20 years. We thank the staff at the NIH Clinical Center and the NEI Eye Clinic for patient phenotyping, ophthalmic testing, and ophthalmic imaging. This work utilized the computational resources of the NIH HPC Biowulf cluster (https://hpc.nih.gov). This study was supported by the Intramural Research Program of the NIH. This research was made possible through access to data in the National Genomic Research Library, which is managed by Genomics England Limited (a company owned by the Department of Health and Social Care). The National Genomic Research Library holds data provided by patients and collected by the NHS as part of their care and data collected as part of their participation in research. GA is supported by a Fight For Sight (UK) Early Career Investigator Award (5045/46), the National Institute of Health Research Biomedical Research Centre (NIHR-BRC) at Moorfields Eye Hospital, and the UCL Institute of Ophthalmology, and Moorfields Eye Charity (Stephen and Elizabeth Archer in memory of Marion Woods) and NIH-P20GM139769. RY was supported by the Moorfields Eye Charity Career Development Award and Springboard (GR001155 and GR001210), Medical Research Council (MR/X001067/1) and FODNECYT (1221843). MIM was supported by Moorfields Eye Charity PhD Studentship (GR001661).

## Author contributions

E.U., R.B.H., M.I.M., G.A., and B.G. performed exome/genome analysis and variant confirmation in the NIH and UK100KGP populations. R.A., A.N., and C.B. performed variant confirmation, including confirmation of deletion breakpoints. I.H.M., D.B., S.L., T.G.T., M.M., and B.P.B. performed patient examination/phenotyping. Y.V.S. performed molecular dynamic modeling. D.M. performed bioinformatic analyses of *NR6A1*. E.B. preformed RNA scope analysis. U.M.N. designed, performed and analyzed zebrafish morpholino knockdown, mRNA rescue experiments and imaging. U.M.N. and D.S.M. performed zebrafish in situ hybridization. R.Y. assisted in interpretation of zebrafish data. A.G. and C.A. performed cell culture and transfection experiments. A.G. performed confocal microscopy. R.V. performed flow cytometry. All co-authors provided draft language and experimental design for their portions of the manuscript, as well as a critical review of the entire manuscript. BPB provided overall project conception and experimental design, drafting of the manuscript and project support.

## Funding

## Competing interests

The authors declare no competing interests.
