## [Transparent Peer Review file · Nature Communications]

Variants in NR6A1 cause a novel Oculo Vertebral Renal syndrome

Corresponding Author: Dr Brian Brooks

Version 0:

Reviewer comments:

Reviewer #1

(Remarks to the Author)

This manuscript addresses the genetic basis of ocular coloboma, a condition of visual impairment. A significant proportion of patients lack a genetic diagnosis, therefore, identifying new genes associated with coloboma is a priority. Here, the authors carry out clinical diagnostic analysis on patients in a National Eye Institute cohort and a UK 100K Genomes Project cohort, and identify a novel syndrome affecting the eye, kidney, and vertebrae, in which the six variants in the orphan nuclear receptor NR6A1 are identified. Gene expression analyses are carried out using existing human data sets, mouse, and zebrafish, and functional experiments are performed using knockdown in zebrafish followed by use of wild type and patient variant forms of nr6a1 to test for rescue.

Overall, this is an exciting description of a new syndrome and the identification of its molecular basis. The information uncovered could aid with future diagnostic testing of coloboma patients. That being said, there are a number of questions regarding the experiments and data as presented in this manuscript. Specific questions and comments are below.

Specific points:

- Fig. 2A, line 169: The R92W variant is described as “forming apparent protein aggregates”. It is not clear that this can be stated from these data (and only one cell is shown); there may be specific subnuclear distribution of this variant form.
- Fig. 3: This analysis is useful, however, it would also be helpful to extend some of the analysis in Fig. 3C to examine expression of NR6A1 in human fetal tissues from the eyeIntegration Project. Fig. 3A appears to involve adult tissues; since some of the phenotypes are considered developmental, expression in human fetal stages would be useful to identify, if possible.
- Line 188-190, “Previous work has demonstrated widespread expression of Nr6a1 in mouse at E8.5 and E9.5 (including the optic vesicle) that becomes nearly undetectable by E12.5.”
This statement needs a citation, especially as the authors do not validate those timepoints themselves.
- Supplementary Fig. 7: Can the authors discuss this expression analysis a bit more? It appears that there is staining in the E11.5 control. I could not find in the methods what the control is and if this is a specific probe.
- Fig. 4: Given the potential expression of nr6a1a in the embryonic eye tissues, including a dorsal view for A and B (11 hpf and 16 hpf) would be extremely useful for evaluating, for example eye tissues compared to periocular tissues.
- Supplementary Fig. 8: Is nr6a1b expressed in a retinal layer at 72 hpf?
- Fig. 5, Supplementary Fig. 9-10, morpholino experiments: Although coloboma is the main focus of the manuscript, did the authors evaluate zebrafish embryos knocked down for nr6a1a and/or nr6a1b for kidney or vertebral defects?
- Fig. 5, Supplementary Fig. 9-10; morpholino rescue experiments: The authors state that they co-inject translation blocking morpholinos with wild type human NR6A1 mRNA. More information would be helpful to evaluate the result. Can they comment on how conserved the morpholino targeting site is in zebrafish compared to human? Did they try using rescuing

the phenotypes by coinjecting the mRNA with the splice blocking morpholinos, which should definitely not affect translation of a rescue construct?

- Fig. 5, Supplementary Fig. 9-10; morpholino rescue experiments: The phenotypic scoring accounts for a variety of phenotypes, many of which are well beyond the coloboma phenotype that initially motivated the study. What percent of embryos were rescued specifically for coloboma (or a kidney or vertebral phenotype, if present)?

Minor points:

- Line 71: The word Mendelian should probably be capitalized.
- Fig. 1H, regarding description of family COL005: It was a bit confusing to realize that the red bar in the figure corresponded to the lesion being described in the text. Could the authors add the designation "p.Ile48Asnfs*3" to the red bar?
- Supplementary Fig. 1A: The text in the figure is very small and difficult to read.
- Fig. 2: Scale bars would be helpful in this figure.
- Fig. 4 Legend: It should be noted that D" and H" are dorsal views (in addition to D' and H').
- Fig. 4 Legend, typo: Tegmentum is misspelled ("tegementum").
- Typo, line 218: "All the morpholinos experiments" should be "morpholino"

Reviewer #2

(Remarks to the Author)

Neelathi and colleagues present a comprehensive study identifying variants in the human NR6A1 gene in individuals with eye, vertebral +/- kidney malformations, with functional studies utilising in vitro subcellular localisation and in vivo morphology to test pathogenicity of 2 novel variants. Eye and vertebral defects align with previous literature on this gene in model organisms. The results are coherent and overall support conclusions, manuscript well written, however I do have concern over the in vivo methodology employed.

I am not an expert on human variant identification and standards for that field and thus will not comment on this section.

Expression analyses:

scRNAseq analysis appropriate and provides support for expression of NR6A1 in tissues relevant to phenotypes. Nice in situ analysis in zebrafish.

Supp Fig 7 – it is difficult to see expression in E10.5 section, can the authors enhance this at all? Please clarify in legend what control is (is this sample processed at all, ie sense transcript [which for this gene would not be good in some genomic regions]), clarify what the signal seen in control is (presumably blood cells?).

Functional analyses:

In vitro subcellular localisation was performed in an overexpression setting, WT localising to the nucleus with variants showing altered localisation, supporting altered protein functionality in patients.

No concerns for this section.

In vivo functional analysis centred on the zebrafish model, where 2 genes exist in the teleost genome. It is unclear why the authors chose a morpholino approach, the field has many concerns with this approach and there is a very straightforward alternative with CRISPR. MO are really only used to complement CRISPR studies. Please see <https://doi.org/10.1371/journal.pgen.1007000>.

Fig 5E, please test for statistical significance – mentioned that neither the hR92W or hR436C NR6A1 mRNAs resulted in significant rescue

There is a curved axis in morphants, but total somite number needs to be quantified given human phenotype.

The following section needs to be deleted:

"We include congenital renal disease as part of this new syndrome not only because two individuals in two separate pedigrees exhibited these phenotypes, but also because Rasouly et al. have simultaneously identified presumed loss-of-function variants in thirteen individuals with congenital renal abnormalities, with or without congenital eye abnormalities, providing further validation of our findings (personal communication)."

It is unclear if this also relates to NR6A1, personal communication is not appropriate, and raises query on the inclusion of kidney defects as part of the syndrome in this manuscript, which need to stand alone.

Reviewer #3

(Remarks to the Author)

Authors describe novel variants from 6 pedigrees in Nr6a1 gene as a cause of autosomal dominant OVR. Evidence to support this includes in silico modeling, cellular localization studies, publicly available expression data, and knock down in zebrafish using morpholinos. Overall strengths include identification and first description of OVR syndrome and missing vertebrate in humans in NEI and UK cohorts. Further strengths include usage of ACMG criteria to classify novel variants into pathogenic or likely pathogenic categories based on evidence presented in this study, and thorough phenotypic data with proper clinical interpretation is valuable for novel gene identification and for ascertaining the phenotypic spectrum. The combination of in vitro mislocalization data, in silico modeling, and zebrafish modeling of variants is an additional strength to support the role of NR6A1 in OVR. This is an impactful discovery as the diagnostic rate of MAC spectrum remains very low, and the description of the first disorder with lacking vertebrae is very important! Together the paper presents significant evidence for NR6A1 variants causing MAC spectrum and OVR. Overall, this is a high impact manuscript and would warrant publication pending addressing the specific areas below:

Major comments:

- 1) Please include additional detail about the rationale for the zebrafish phenotypes and experiments that were used to assess rescue. While microphthalmia and optic fissure closure are phenotypes associated with OVR, it is necessary to document the relevance of bent body axis and heart edema. The phenotypes relative to renal and heart defects and vertebrae should be assessed in the zebrafish if possible. Is the number of vertebrae affected in the morpholino knockdown for example. Additionally, several details in the experiments could be expanded to be clear for non-zebrafish audience, i.e. the use of p53 morpholino and how that shows widespread cell death is not the primary cause; rationale for use of translation and splice blocking morpholinos;
- 2) Conclusions from western blot data to support equal protein expression (supplemental figure 6) are misleading, as there appears to be higher levels of R92W expression in these experiments, though this may not be statistically significant. The flow cytometry data support similar levels of expression. These experiments may need to be redone with more n to support that the results of localization experiments are not due to variable levels of transfection.
- 3) For the localization experiments, quantitative analysis would be helpful for the existing images/experiments as there are some rare aggregates in the wild-type NR6A1. This would help substantiate the consistency of the phenotype among replicates and could be done with existing images.
- 4) Some of the conclusions from the in silico modeling are overstated without further functional experiments to support these strong claims, so these either need to be tempered or additional evidence provided to support these claims. For example, there is a claim that R436C could affect ligand binding, but this is not tested. Additionally, there is significant modeling evidence to support that R92W may impact DNA binding, but functional evidence is not shown. EMSA would be very helpful to assess the DNA binding ability of R92W.
- 5) RNAscope data in mouse are not very convincing as there is signal outside of areas with cells. The control shows autofluorescence signal, so this extra channel should be included for each of the timepoints. Comparing signal to other timepoints or areas of the embryo where this is no expression would be helpful i.e. E9.5 or neighboring brain tissue to be able to adequately assess the specificity. Can the authors comment on discrepancy between mouse and zebrafish expression pattern? What is the role of the lens in coloboma, if anything?
- 6) Nonsense mediated decay is listed as a mechanism for truncating variants, however, this is not fully explored. Looking for stable mRNA transcripts in blood/saliva samples that were taken to address the nonsense mediated decay would be helpful where RNA samples are available.

Minor comments:

- 1) Line 64-65: Mentions Nr6a1's role in stem cell maintenance in the introduction but does not revisit this in the discussion or anywhere in the paper
- 2) Line 71-72: Interesting that this has never been described in humans and a lot of data is presented in the discussion demonstrating this has been seen a lot in other organisms... why is this so rare in humans? This may be a noteworthy point to address in the discussion.
- 3) Line 106: Can the presence of kidney be confirmed with clinical records or is only family historical data available?
- 4) Line 157: This sentence needs to be clarified. Distance of residues are listed for the variant domain but does not also state the distance of these residues in the WT. Also a study or rationale need to be provided to support the conclusion that the cysteine is protected, and the statement about oxidative damage is very speculative and should be removed.
- 5) Line 166: Western blot from supplement shows almost a two-fold difference in protein levels with one outlier likely making it statistically insignificant - please address with additional replicates as noted above.
- 6) Line 167-169: R92W variant forms protein aggregates in the nucleus which appears to coincide exactly within the nucleoli. Can the authors comment on this?
- 7) Line 181-184 and line 327-328: Discusses the correlation between coloboma associated genes in fetal ocular tissues - I do not understand how this was done even after reading the methods and looking at the figure caption so further clarification of the methods are needed.
- 8) Line 243-245: This sentence is needs to be rephrased to be less confusing – keep consistent comparison – either compare % normal or % severe/moderate in the text. The figure makes it clearer.
- 9) Line 250-257: What happens to zebrafish if you inject up to 200pg of a non-targeting mRNA – nonspecific control should be included here to show the specificity of overexpression or injection of nonfunctional mutant RNA.
- 10) Line 291-295: Insufficient evidence is presented in the manuscript to draw conclusions on disease mechanism – haploinsufficiency vs. dominant negative vs. dominant toxicity and based on the localization studies the authors suggest that multiple mechanisms may be at play. I'd remove the speculations about mechanisms or include arguments that certain mechanisms. i.e. Loss of mRNA or the lack of stable produced protein would suggest haploinsufficiency as a mechanism.
- 11) Line 327-328: Cite the previous paper for the mouse expression - current data provided with RNAscope is not sufficient

or convincing on a definitive expression pattern in mouse

Reviewer #4

(Remarks to the Author)

Version 1:

Reviewer comments:

Reviewer #1

(Remarks to the Author)

The revised version of this manuscript is significantly improved: the authors have responded to the questions and comments raised in the prior round of review. There is one minor comment below.

- Supplementary Fig. 10: It is appreciated that the dorsal views are included in the revised manuscript. One minor point is that the optic cup is outlined with a dotted line at 16 hpf. The optic cup has not undergone invagination at this time, therefore, it should not be a cup shape as drawn.

Reviewer #2

(Remarks to the Author)

All points have been appropriately addressed in this high quality manuscript. This includes revised mouse section analysis and analysis of somite morphology and number. Morpholino experiments have been controlled as best as they can, and align well with data in complementary manuscripts online now. I have no further concerns.

Reviewer #3

(Remarks to the Author)

The current study is an excellent addition to the literature and establishes NR6A1 as a new gene for coloboma and is acceptable for publication following fixing of minor areas below. The authors addressed all of our initial concerns, and concerns about the morpholino approach for validation. The additional data from two other groups also helps to corroborate many of the conclusions from this study. There were a few minor changes that need to be corrected:

- 1) Line 896, missing period: "expressionThe "
- 2) The RNAscope data for NR6A1 in mice was still not the most convincing, but in the context of corroborating data regarding expression pattern this is acceptable.
- 3) SupFig9 has a typo "Brian vesicle" I imagine this should be "brain vesicle".
- 4) For the human fetal tissue expression data, only Mellough et al include time points that are relevant to optic fissure closure. As such, it may be prudent just to include this data set in Figure 3. The other two fetal datasets are irrelevant as they are at later times.
- 5) "The same data used in the plotting was used to create separate gene expression matrices that for 894 either fetal primary tissue (retina and RPE) or adult tissue (retina and RPE). We used spatial quantile 895 normalization (spqn) analysis to derive gene correlation matrices that were not biased by higher gene 896 expression" – this text is duplicated and should be edited.

Reviewer #4

(Remarks to the Author)

We are delighted to submit revisions to our manuscript, “**Variants in *NR6A1* cause a novel oculo-vertebral-renal (OVR) syndrome,**” (NCOMMS-24-72070-T) for your consideration. We very much appreciate the comments of our reviewers and feel that they have helped to improve the manuscript. We have pasted the reviews below and given a point-by-point response.

Reviewer #1 (Remarks to the Author):

This manuscript addresses the genetic basis of ocular coloboma, a condition of visual impairment. A significant proportion of patients lack a genetic diagnosis, therefore, identifying new genes associated with coloboma is a priority. Here, the authors carry out clinical diagnostic analysis on patients in a National Eye Institute cohort and a UK 100K Genomes Project cohort, and identify a novel syndrome affecting the eye, kidney, and vertebrae, in which the six variants in the orphan nuclear receptor *NR6A1* are identified. Gene expression analyses are carried out using existing human data sets, mouse, and zebrafish, and functional experiments are performed using knockdown in zebrafish followed by use of wild type and patient variant forms of *nr6a1* to test for rescue.

Overall, this is an exciting description of a new syndrome and the identification of its molecular basis. The information uncovered could aid with future diagnostic testing of coloboma patients. That being said, there are a number of questions regarding the experiments and data as presented in this manuscript. Specific questions and comments are below.

Specific points:

1. Fig. 2A, line 169: The R92W variant is described as “forming apparent protein aggregates”. It is not clear that this can be stated from these data (and only one cell is shown); there may be specific subnuclear distribution of this variant form.

We agree with this distinction. As such, we have revised the language to describe the signal as “not uniformly distributed across the nucleus”, which could cover both the possibility of aggregates and of localization to specific nuclear sub-structures. We have also now performed an immunofluorescence experiment using a nucleolus marker FIBRILLARIN, showing that the mutant R92W protein signal appears to be distinct from the nucleolus (Supplementary Fig. 7). We manually counted and analyzed 350 cells with R92W-GFP expression, and the observed localization pattern was similar in all cases.

2. Fig. 3: This analysis is useful, however, it would also be helpful to extend some of the analysis in Fig. 3C to examine expression of NR6A1 in human fetal tissues from the eyeIntegration Project. Fig. 3A appears to involve adult tissues; since some of the phenotypes are considered developmental, expression in human fetal stages would be useful to identify, if possible.

We agree that human embryonic data are more useful than data from adult tissue in this context. Optic fissure closure in humans occurs around 5 weeks post-conception and much of the human fetal data are for time points after that. However, we have revised Fig. 3 to include a time course of bulk RNA-Seq data from three studies reporting on human fetal tissue (panel b). We have inserted a new sentence, “Consistent with this observation, bulk RNA-Seq data from human fetal tissue shows that NR6A1 expression is highest in early stages of development, including the time of optic fissure closure in the first trimester (Fig. 3b).” The figure legend has been updated to read, “Bulk RNA-Seq data in human fetal tissue from three studies suggests NR6A1 expression is highest in early stages of development, including the window of optic fissure closure (lavender box). NR6A1 expression is plotted against the tissue age (days post conception, dpc). A linear regression analysis was added for each paper’s data from the 40 to 80 dpc and 80 to 160 dpc samples.” We have removed the Human Retinal Cell Atlas Data from the main figure and made this point in a supplemental figure (Supplementary Fig. 8).

3. Line 188-190, “Previous work has demonstrated widespread expression of Nr6a1 in mouse at E8.5 and E95 (including the optic vesicle) that becomes nearly undetectable by E12.5.” This statement needs a citation, especially as the authors do not validate those timepoints themselves.

We appreciate the reviewer pointing this out. We have added the appropriate citation: Chang YC et al. from PMID: 36522318.

4. Supplementary Fig. 7: Can the authors discuss this expression analysis a bit more? It appears that there is staining in the E11.5 control. I could not find in the methods what the control is and if this is a specific probe.

Please note that Supplementary Fig. 7 is now Supplementary Fig. 9. We appreciate this point and agree the figure and its associated text could be clearer. Reviewer #2 made a similar point about this figure. With RNA-Scope we occasionally observe what appears to be variable

amounts of autofluorescence in erythrocytes and this is the signal the reviewer is noting in our control. In reviewing all the images we have, we realized that this autofluorescence was NOT observed in the vast majority of samples, making the original panel not representative of the whole. (These same images also did NOT have appreciable signal in the optic cup proper.). We have corrected the figure to include a more representative section that does not show this signal and hope that this is less confusing to the readership. We have also clarified in the text that notes, "The manufacturer's control probe against a bacterial sequence was used for reference", and updated in the Supplemental methods adding "(Cat# 320871, Advanced Cell Dianogstics)". A similar sentence has been placed in the figure legend. We have also expanded the methods for this section as requested. We also ask that the reviewer view the original image, as we note that the image within the in line text is not as high of quality.

5. Fig. 4: Given the potential expression of *nr6a1a* in the embryonic eye tissues, including a dorsal view for A and B (11 hpf and 16 hpf) would be extremely useful for evaluating, for example eye tissues compared to periocular tissues.

*The reviewer makes an excellent point. To help the reader distinguish between ocular and non-ocular expression of *nr6a1a*, we have now included Supplementary Fig. 10 to show the dorsal view at 11hpf and 16hpf. The dorsal view at 11hpf substantiates the broad expression, while this view at 16hpf demonstrates that the ocular expression is lower than that of the surrounding brain tissue. We note this in the text by adding, "expression in the developing eye is reduced compared to the adjacent developing brain."*

6. Supplementary Fig. 8: Is *nr6a1b* expressed in a retinal layer at 72 hpf?

We have now included a higher magnification image alongside a sense probe image to help clarify this. The new figure number is Supplementary Figure 12. We note that there is faint expression in the retina and likely in the presumed RPE. This is now noted in the text and the figure legend (Supplementary Fig. 12d, e).

7. Fig. 5, Supplementary Fig. 9-10, morpholino experiments: Although coloboma is the main focus of the manuscript, did the authors evaluate zebrafish embryos knocked down for *nr6a1a* and/or *nr6a1b* for kidney or vertebral defects?

*We agree with the reviewer and these comments echo those of reviewer #3. To help address these concerns, we have now included a new figure (Fig. 6) that includes data detailing the expression of *nphs1* and *nphs2* (two markers of kidney development) in control and morpholino (*nr6a1a+nr6a1b*) experiments. We note that many embryos have reduced or nearly absent expression of both markers at 48 hpf. Interestingly, some embryos showed asymmetric changes in expression, consistent with the human phenotype of renal agenesis, where one of the kidneys is missing.*

To address the vertebral/segmentation defects, we have included in Fig.6, a quantification of the number of somites in control vs. the nr6a1a+nr6a1b double morphant embryos at 24 hpf, showing a significant decrease in the latter. Somites are the precursors from which the sclerotome (vertebral precursors) are formed. Furthermore, we have observed patchy and/or reduced expression of pax9, a marker for the ventromedial region of somites (sclerotome) from which vertebrae are formed. Lastly, Jacquinet et al. have recently reported that stable nr6a1a and nr6a1b mutants demonstrate a reduced number of vertebrae, in agreement with our observations and with the patient phenotype (Jacquinet, Adeline, et al. "Variants in NR6A1 as a cause for congenital renal, vertebral and uterine anomalies." medRxiv (2025): 2025-01.)

8. Fig. 5, Supplementary Fig. 9-10; morpholino rescue experiments: The authors state that they co-inject translation blocking morpholinos with wild type human NR6A1 mRNA. More information would be helpful to evaluate the result. Can they comment on how conserved the morpholino targeting site is in zebrafish compared to human? Did they try using rescuing the phenotypes by coinjecting the mRNA with the splice blocking morpholinos, which should definitely not affect translation of a rescue construct?

The human mRNA with which we are rescuing the nr6a1(a+b) morphant phenotype, does not have any overlap with the MO targeting sites of either nr6a1a-MO or nr6a1b-MO. We have provided a Clustal alignment as Supplementary Fig. 13. To clarify this in the text, we have inserted the following sentence: "The sequence of the TB morpholinos does not significantly overlap with the human mRNA sequence and is therefore unlikely to interfere with mRNA rescue experiments." A figure legend stating the point is also included.

9. Fig. 5, Supplementary Fig. 9-10; morpholino rescue experiments: The phenotypic scoring accounts for a variety of phenotypes, many of which are well beyond the coloboma phenotype that initially motivated the study. What percent of embryos were rescued specifically for coloboma (or a kidney or vertebral phenotype, if present)?

The reviewer raises a good point. We have now addressed this, where we show approximately 55% rescue for coloboma, 53% for body axis and 44% for heart edema. This is noted now in the text: "Breaking down each phenotype separately, we note that we are able to rescue approximately 55% embryos for coloboma, 53% for body axis and 44% for heart edema."

Minor points:

- Line 71: The word Mendelian should probably be capitalized.
Mendelian has been capitalized.

- Fig. 1H, regarding description of family COL005: It was a bit confusing to realize that the red bar in the figure corresponded to the lesion being described in the text. Could the authors add the designation "p.Ile48Asnfs*3" to the red bar?

Figure has been updated as suggested

- Supplementary Fig. 1A: The text in the figure is very small and difficult to read.

Font size increased.

- Fig. 2: Scale bars would be helpful in this figure.

Scale bars have been added in this figure

- Fig. 4 Legend: It should be noted that D'' and H'' are dorsal views (in addition to D' and H').

The legend has been updated.

- Fig. 4 Legend, typo: Tegmentum is misspelled ("tegementum").

Spelling corrected

- Typo, line 218: "All the morpholinos experiments" should be "morpholino"

Correction incorporated

Thank you for noting these. These changes have been made as advised.

Reviewer #2 (Remarks to the Author):

Neelathi and colleagues present a comprehensive study identifying variants in the human NR6A1 gene in individuals with eye, vertebral +/- kidney malformations, with functional studies utilising in vitro subcellular localisation and in vivo morphology to test pathogenicity of 2 novel variants. Eye and vertebral defects align with previous literature on this gene in model organisms. The results are coherent and overall support conclusions, manuscript well written, however I do have concern over the in vivo methodology employed.

I am not an expert on human variant identification and standards for that field and thus will not comment on this section.

1. Expression analyses: scRNAseq analysis appropriate and provides support for expression of NR6A1 in tissues relevant to phenotypes. Nice in situ analysis in zebrafish.

Thank you.

2. Supp Fig 7 – it is difficult to see expression in E10.5 section, can the authors enhance this at all? Please clarify in legend what control is (is this sample processed at all, ie sense transcript [which for this gene would not be good in some genomic regions]), clarify what the signal seen in control is (presumably blood cells?).

The reviewer makes several important points, similar to those raised by reviewer #1. Please see response above, including clarification that the control is a manufacturer-supplied probe against an unrelated bacterial sequence. We also note that the signal noted is likely autofluorescence from erythrocytes, which actually is NOT seen in most sections. This is now corrected. Because the signal is indeed faint, we recommend viewing the original image as the pasted image within the in line text is not as high quality. We have also added some more information on our methodology to highlight how images were captured and processed.

3. Functional analyses: In vitro subcellular localisation was performed in an overexpression setting, WT localising to the nucleus with variants showing altered localisation, supporting altered protein functionality in patients. No concerns for this section.

Thank you.

4. In vivo functional analysis centered on the zebrafish model, where 2 genes exist in the teleost genome. It is unclear why the authors chose a morpholino approach, the field has many concerns with this approach and there is a very straightforward alternative with CRISPR. MO are really only used to complement CRISPR studies. Please see <https://doi.org/10.1371/journal.pgen.1007000>.

We appreciate this comment and thank the reviewer for noting this reference, as we had originally cited it, but that citation appears to have been accidentally deleted in the editing process.

Indeed, the advent of CRISPR has made the generation of mutant fish easier than in the past, although creation of a stable line is a considerably longer process that comes with its own caveats. Furthermore, multiple studies, included one cited in the PLoS Genetics guidelines in 2017, have found that one reason that CRISPR mutants often have milder or no phenotypes compared to morpholino experiments is because of genetic compensation (Jakutis and Stainier, 2021); as such, they may not recapitulate a human phenotype even in the presence of strong evidence of the involvement of a gene as a Mendelian cause of disease. In fact, many of the authors of the reference paper cited, still publish results using morpholinos, reinforcing that these are still a valid and current approach for gene knockdown.

We chose the morpholino approach to circumvent the genetic compensation issue and followed the guidance provided in the Stanier et al., 2017 reference and have now noted this explicitly in the new version of our manuscript:

“All MO experiments were carried out following the guidelines set forth for their use in zebrafish³¹⁻³³. These guidelines include: 1) use of two non-overlapping MOs (one translation blocking (TB), one splice blocking (SB)); 2) observation of a consistent phenotype with both TB and SB MOs for each paralog; 3) a correlation between dose of MO and phenotype, with lower

concentrations of MO causing a milder phenotype; 4) validation of the efficacy of SB MOs by RT-PCR analysis; 5) lack of a phenotype with injection of a control MO; and; 6) partial rescue of the MO phenotype with co-injection of the corresponding human mRNA”.

Given our careful adherence to these guidelines, the consistency of the zebrafish phenotype with that of humans—especially in the revised manuscript with kidney and somite marker studies—and the decreased ability of mutant human mRNA to rescue the morpholino phenotype compared to wild-type mRNA, we argue that our experiments are scientifically rigorous and fulfil the expected conditions and outcomes stated in the reference guidelines, even in the CRISPR era.

Lastly, while we were pursuing further experiments for this revised manuscript, Jacquinet et al. submitted a preprint (<https://www.medrxiv.org/content/10.1101/2025.01.08.24319478v1>) describing the phenotype of zebrafish CRISPR lines for both nr6a1a and nr6a1b. They note their fish show renal development and vertebral segmentation abnormalities, similar to our morpholino experiments and to our patients’ phenotype. We feel these observations provide external validation to our work.

5. Fig 5E, please test for statistical significance – mentioned that neither the hR92W or hR436C NR6A1 mRNAs resulted in significant rescue. There is a curved axis in morphants, but total somite number needs be quantified given human phenotype.

We agree. As such, we have now included a new figure (Fig. 6) where we show abnormal somite morphology, abnormal expression of the sclerotome marker pax9, and quantitated a reduced number of somites (Fig. 6s). We have also tested the statistical significance and have updated the in the figures and the figure legends.

6. The following section needs to be deleted: "We include congenital renal disease as part of this new syndrome not only because two individuals in two separate pedigrees exhibited these phenotypes, but also because Rasouly et al. have simultaneously identified presumed loss-of-function variants in thirteen individuals with congenital renal abnormalities, with or without congenital eye abnormalities, providing further validation of our findings (personal communication)." It is unclear if this also relates to NR6A1, personal communication is not appropriate, and raises query on the inclusion of kidney defects as part of the syndrome in this manuscript, which need to stand alone.

This is a fair point. We have now updated this statement with a citation to reference a preprint: Milo Rasouly, Hila, et al. "Exome-wide analysis of congenital kidney anomalies reveals new genes and shared architecture with developmental disorders." medRxiv (2024): 2024-11. Approximately two months after the submission of our article to medRxiv, as noted above yet a third group has found variants in NR6A1 in patients with congenital uterine and renal anomalies, along with missing vertebrae: Jacquinet, Adeline, et al. "Variants in NR6A1 as a cause for congenital renal, vertebral and uterine anomalies." medRxiv (2025): 2025-01. As

such, we have also included a sentence in the discussion to mention this. We have confirmed with the editors at Nature Communications that citation of preprints is consistent with journal policy.

Reviewer #3 (Remarks to the Author):

Authors describe novel variants from 6 pedigrees in Nr6a1 gene as a cause of autosomal dominant OVR. Evidence to support this includes in silico modeling, cellular localization studies, publicly available expression data, and knock down in zebrafish using morpholinos. Overall strengths include identification and first description of OVR syndrome and missing vertebrate in humans in NEI and UK cohorts. Further strengths include usage of ACMG criteria to classify novel variants into pathogenic or likely pathogenic categories based on evidence presented in this study, and thorough phenotypic data with proper clinical interpretation is valuable for novel gene identification and for ascertaining the phenotypic spectrum. The combination of in vitro mislocalization data, in silico modeling, and zebrafish modeling of variants is an additional strength to support the role of NR6A1 in OVR. This is an impactful discovery as the diagnostic rate of MAC spectrum remains very low, and the description of the first disorder with lacking vertebrae is very important! Together the paper presents significant evidence for NR6A1 variants causing MAC spectrum and OVR. Overall, this is a high impact manuscript and would warrant publication pending addressing the specific areas below:

Major comments:

1) Please include additional detail about the rationale for the zebrafish phenotypes and experiments that were used to assess rescue. While microphthalmia and optic fissure closure are phenotypes associated with OVR, it is necessary to document the relevance of bent body axis and heart edema. The phenotypes relative to renal and heart defects and vertebrae should be assessed in the zebrafish if possible. Is the number of vertebrae affected in the morpholino knockdown for example. Additionally, several details in the experiments could be expanded to be clear for non-zebrafish audience, i.e. the use of p53 morpholino and how that shows widespread cell death is not the primary cause; rationale for use of translation and splice blocking morpholinos;

We appreciate this excellent comment, which touches on a point raised by Reviewer #1. Given that both vertebral and renal phenotypes are part of the syndrome we describe, doing at least a preliminary assessment of these organs is important. Please see a detailed explanation of the additional data we have provided regarding the renal and vertebral phenotype of the zebrafish embryos above. In short, we have shown that the patterns of two renal markers are altered in the morphant zebrafish, that somite morphology is altered, that the sclerotome marker pax9 is altered, and that the number of somites is reduced (Fig. 6). (Somites, as the reviewer likely knows, are the precursors of the vertebrae with the sclerotome being the specific subdivision of the somites that leads to vertebrae.) Furthermore, as noted by reviewers #1 and #2, the preprint by Jacques et al., 2025 Medrxiv shows that nr6a1a and nr6a1b mutants have fewer vertebrae.

The reviewer is correct in pointing out that the morphants' heart edema was not thoroughly discussed. Although none of the patients we report have a history of congenital heart problems, the reviewer is also correct in pointing out that the heart primordium does express nr6a1a, which is now highlighted in a Fig. 4b, Supplementary Fig. 11. The heart edema we observe in nr6a1a morphants is consistent with the transient heart edema observed in nr6a1a mutants in a preprint submitted subsequent to our report (Jacquinet et al., 2025 Medrxiv). Therefore, we have added the following sentence to the discussion to point out that, at present, we do not include the heart as part of this syndrome, but the subsequent description of other patients with this disorder could very well expand the phenotypic spectrum: "Because none of our affected patients displayed heart defects, we currently do not include this as part of the syndrome, even though heart edema was noted in the morphants. However, we cannot rule out that congenital heart anomalies will be found in subsequent patients as more are identified."

The rationale in using translational, splice block morpholinos and p53 morpholinos are described in detail in answers to the reviewer 2 comments; these are also incorporated in the text..

2) Conclusions from western blot data to support equal protein expression (supplemental figure 6) are misleading, as there appears to be higher levels of R92W expression in these experiments, though this may not be statistically significant. The flow cytometry data support similar levels of expression. These experiments may need to be redone with more n to support that the results of localization experiments are not due to variable levels of transfection.

The reviewer is correct in pointing out that the R92W protein has higher levels of expression, even though this was not statistically significant. However, the point of this western blot was not to show relative levels of protein expression; rather, it was to demonstrate that the transfection of the different NR6A1 variants leads to the expression of a protein product of the expected molecular weight. Because we think that the relative levels were beside the point we were trying to make, we have not performed additional experiments at this time.

3) For the localization experiments, quantitative analysis would be helpful for the existing images/experiments as there are some rare aggregates in the wild-type NR6A1. This would help substantiate the consistency of the phenotype among replicates and could be done with existing images.

We note in the figure legend that "The localization pattern for the WT and the two variant isoforms was observed to be consistent across three transfection experiments. (Cells counted: WT=387, R92W=350 and R436C=217)." We have also provided a low magnification image of each protein to demonstrate that the localization pattern was consistent over multiple cells.

4) Some of the conclusions from the in silico modeling are overstated without further functional experiments to support these strong claims, so these either need to be tempered or additional evidence provided to support these claims. For example, there is a claim that R436C could affect

ligand binding, but this is not tested. Additionally, there is significant modeling evidence to support that R92W may impact DNA binding, but functional evidence is not shown. EMSA would be very helpful to assess the DNA binding ability of R92W.

The reviewer makes a good point. We have now revised the text to only include what the model shows, rather than to speculate on the functional effect. Others have noted a putative ligand binding domain and so our statement for the R92W variant is only noting that this falls in that region of the protein.

5) RNAscope data in mouse are not very convincing as there is signal outside of areas with cells. The control shows autofluorescence signal, so this extra channel should be included for each of the timepoints. Comparing signal to other timepoints or areas of the embryo where this is no expression would be helpful i.e. E9.5 or neighboring brain tissue to be able to adequately assess the specificity. Can the authors comment on discrepancy between mouse and zebrafish expression pattern? What is the role of the lens in coloboma, if anything?

These points are well taken. We interpret the signal from E10.5 embryos as low level expression, in that it is above that of E11.5 and of controls. This low level of expression appears to be fairly generalized and approximately that of what is observed in brain. To help illustrate this, we have revised Supplementary Figure 7, which is now Supplementary Fig. 9 to include a low magnification image of the developing eye plus brain, with quantitation of the expression in these two areas at both E10.5 and E11.5. We also note that these levels are generally higher than that observed in at least some areas of surrounding mesenchyme. The signal that appears outside the cells likely represents signal “bleed through” from cells that are beneath the plane of focus, as we do not see nonspecific signal in areas that are clearly outside of the main tissue slice. We hope that the lower magnification photo where we show brain tissue for comparison will help to illustrate this point.

In performing these additional experiments, we noted that lens expression was occasionally noted in E10.5 embryos, which may be an equivalent to the nr6a1b expression observed in zebrafish. As such, we have deleted the phrase that draws this distinction. Both cell autonomous and non-cell autonomous factors—including those from the lens—are known to regulate optic cup morphogenesis and optic fissure closure. We include a sentence in the discussion to point out that either or both mechanisms are possible for coloboma and give one example related to a cavefish model of microphthalmia.

6) Nonsense mediated decay is listed as a mechanism for truncating variants, however, this is not fully explored. Looking for stable mRNA transcripts in blood/saliva samples that were taken to address the nonsense mediated decay would be helpful where RNA samples are available.

This would be a wonderful experiment to perform. However, we currently do not have access to patient cells that would permit us to look at mRNA steady state levels.

Minor comments:

1) Line 64-65: Mentions Nr6a1's role in stem cell maintenance in the introduction but does not revisit this in the discussion or anywhere in the paper.

We have now added the following sentence to the discussion to mention the role of Nr6a1 in stem cell biology: "Given the known role of Nr6a1 in stem cell biology, we posit that the developmental trajectory of the optic cup neuroepithelium is altered in a way not consistent with optic fissure closure." Speaking in general terms, the developmental program of optic fissure closure likely involves a complex sequence of molecular and cellular events and that if something perturbs these, the "default" state of the optic fissure is to remain open.

(2) Line 71-72: Interesting that this has never been described in humans and a lot of data is presented in the discussion demonstrating this has been seen a lot in other organisms... why is this so rare in humans? This may be a noteworthy point to address in the discussion.

This is also an interesting point. It is purely speculative, but it is possible, given the importance of NR6A1 in early stages of development that embryos/fetuses with pathogenic variants may often spontaneously abort and therefore are unrecognized. Coloboma is also a rare condition (~1/10,000 live births) with considerable genetic heterogeneity. As such, the timing of this discovery may be more of an issue of affordability of exome/genome sequencing for large numbers of patients. To our knowledge, this is indeed the first human disorder where missing vertebrae is a consistent phenotype. However, this would not be easily ascertained if spine x-rays are not consistently ordered on all patients with coloboma. This latter point is now included in the discussion, paragraph 2. Lastly, the penetrance of developmental phenotypes is almost certainly modified by other genetic variants in the genome in a way that can vary from patient to patient.

3) Line 106: Can the presence of kidney be confirmed with clinical records or is only family historical data available?

We do not have access to individual COL005.17 at present. However, for individual COL034.1, we have a prenatal scan report showing a missing kidney which was discussed in Extended Data and we have confirmed the kidney phenotype by renal ultrasound when the proband visited NIH. Given the rarity and specificity of renal agenesis, we submit that the reported phenotype for COL005.17 is indeed likely correct and related. Furthermore, both Rasouly et al and Jacquinet et al have noted renal malformations in their patients, making us more confident that this is indeed part of the OVR phenotype.

4) Line 157: This sentence needs to be clarified. Distance of residues are listed for the variant domain but does not also state the distance of these residues in the WT. Also a study or rationale need to be provided to support the conclusion that the cysteine is protected, and the statement about oxidative damage is very speculative and should be removed.

The reviewer is correct that this is speculation and not experimentally investigated. As such, these statements have been removed. The distances between the cysteine residues in the wild type has been added in the text which reads “The variant R436C breaks this bond and creates a cysteine residue which could form abnormal disulfide bridges in the variant protein, since residues C443, C391, and C422 are distanced at 8-12 Å from C436 in this variant domain as compared to 14-19 Å (C443-C391), 11.09 Å (C443-C422) and 4.52 Å (C91-C422) in the WT protein model”

5) Line 166: Western blot from supplement shows almost a two-fold difference in protein levels with one outlier likely making it statistically insignificant - please address with additional replicates as noted above.

See comment above. The main point of the Western blot is to show that our constructs express a protein of an expected molecular weight, not really to compare relative levels of expression. We don't think we can confidently comment on what the steady state levels of expression would be

6) Line 167-169: R92W variant forms protein aggregates in the nucleus which appears to coincide exactly within the nucleoli. Can the authors comment on this?

Reviewer #1 also noted this. Aggregates may be too specific of a term. As such we have noted simply that the signal is not “uniformly distributed in the nucleus.” Furthermore, we have shown that this signal does NOT appear to colocalize with nucleoli. Please see Supplementary Fig. 9.

7) Line 181-184 and line 327-328: Discusses the correlation between coloboma associated genes in fetal ocular tissues - I do not understand how this was done even after reading the methods and looking at the figure caption so further clarification of the methods are needed.

Thank you for the comment, we have described the correlation in detail in the Supplemental Material and Methods sections with sub heading “Correlation analysis of NR6A1”

8) Line 243-245: This sentence needs to be rephrased to be less confusing – keep consistent comparison – either compare % normal or % severe/moderate in the text. The figure makes it clearer.

This sentence has been updated in the text to read: “nr6a1(a+b), resulted in 16% and 49% embryos having severe and moderate phenotypes respectively”

9) Line 250-257: What happens to zebrafish if you inject up to 200pg of a non-targeting mRNA – nonspecific control should be included here to show the specificity of overexpression or injection of nonfunctional mutant RNA.

To help address the point of possible mRNA toxicity, we have injected 200pg of a CAAX-GFP mRNA, showing that the embryo morphology is well preserved. Please see Supplementary Fig. 16g-h’).

10) Line 291-295: Insufficient evidence is presented in the manuscript to draw conclusions on disease mechanism – haploinsufficiency vs. dominant negative vs. dominant toxicity and based on the localization studies the authors suggest that multiple mechanisms may be at play. I’d remove the speculations about mechanisms or include arguments that certain mechanisms. i.e. Loss of mRNA or the lack of stable produced protein would suggest haploinsufficiency as a mechanism.

We have now updated this paragraph to read, “Additional studies are required to understand the detailed disease mechanisms of NR6A1 variants. Deletion and presumed truncating variants are generally associated with a haploinsufficiency mechanism such as nonsense-mediated decay. This may be tested by expression profiling in patient cells. However, the subcellular localization defects of the two missense variants in NR6A1 hint at more than one mechanism of disease”

11) Line 327-328: Cite the previous paper for the mouse expression - current data provided with RNAscope is not sufficient or convincing on a definitive expression pattern in mouse

Citation added. We cited the work of Chang YC et al. from PMID: 36522318.

Reviewer #4 (Remarks to the Author):

Thank you for taking the time to co-review this manuscript.

We hope that our responses and the additional experiments we have performed sufficiently address the reviewers points and look forward to your further thoughts.

Reviewer #1 (Remarks to the Author):

The revised version of this manuscript is significantly improved: the authors have responded to the questions and comments raised in the prior round of review. There is one minor comment below.

- Supplementary Fig. 10: It is appreciated that the dorsal views are included in the revised manuscript. One minor point is that the optic cup is outlined with a dotted line at 16 hpf. The optic cup has not undergone invagination at this time, therefore, it should not be a cup shape as drawn.

Thank you for the time to review our manuscript, we have removed the markings around the optic vesicle.

Reviewer #2 (Remarks to the Author):

All points have been appropriately addressed in this high quality manuscript. This includes revised mouse section analysis and analysis of somite morphology and number. Morpholino experiments have been controlled as best as they can and align well with data in complementary manuscripts online now. I have no further concerns.

Thank you for the time to review our manuscript.

Reviewer #3 (Remarks to the Author):

The current study is an excellent addition to the literature and establishes NR6A1 as a new gene for coloboma and is acceptable for publication following fixing of minor areas below. The authors addressed all of our initial concerns, and concerns about the morpholino approach for validation. The additional data from two other groups also helps to corroborate many of the conclusions from this study. There were a few minor changes that need to be corrected:

- 1) Line 896, missing period: “expressionThe “

Period has been included between the words.

2) The RNAscope data for NR6A1 in mice was still not the most convincing, but in the context of corroborating data regarding expression pattern this is acceptable.

Thank you.

3) SupFig9 has a typo “Brian vesicle” I imagine this should be “brain vesicle”.

Corrected.

4) For the human fetal tissue expression data, only Mellough et al include time points that are relevant to optic fissure closure. As such, it may be prudent just to include this data set in Figure 3. The other two fetal datasets are irrelevant as they are at later times.

We have removed Aldiri et al., from the analysis as suggested by the reviewer, however we still used Hoshino et al., along with Mellough et al., because they both show a similar trend in decreasing NR6A1 eye expression in early human development.

5) “The same data used in the plotting was used to create separate gene expression matrices that for 894 either fetal primary tissue (retina and RPE) or adult tissue (retina and RPE). We used spatial quantile 895 normalization (spqn) analysis to derive gene correlation matrices that were not biased by higher gene 896 expression” – this text is duplicated and should be edited.

This sentence has been rephrased.

Reviewer #4 (Remarks to the Author):

Thank you for taking the time to co-review this manuscript.

We hope that our responses sufficiently address the reviewers' points.